# Cyclin F-dependent degradation of E2F7 is critical for DNA repair and G2-phase progression

Ruixue Yuan[1,†] (ID), Qingwu Liu[1,†], Hendrika A Segeren[1] (ID), Laurensia Yuniati[2], Daniele Guardavaccaro[2,3], Robert J Lebbink[4], Bart Westendorp[1,*] (ID) & Alain de Bruin[1,5,**] (ID)

## Abstract

**E2F7 and E2F8 act as tumor suppressors via transcriptional repression of genes involved in S-phase entry and progression. Previously, we demonstrated that these atypical E2Fs are degraded by APC/C$^{Cdh1}$ during G1 phase of the cell cycle. However, the mechanism driving the downregulation of atypical E2Fs during G2 phase is unknown. Here, we show that E2F7 is targeted for degradation by the E3 ubiquitin ligase SCF$^{cyclin\ F}$ during G2. Cyclin F binds via its cyclin domain to a conserved C-terminal CY motif on E2F7. An E2F7 mutant unable to interact with SCF$^{cyclin\ F}$ remains stable during G2. Furthermore, SCF$^{cyclin\ F}$ can also interact and induce degradation of E2F8. However, this does not require the cyclin domain of SCF$^{cyclin\ F}$ nor the CY motifs in the C-terminus of E2F8, implying a different regulatory mechanism than for E2F7. Importantly, depletion of cyclin F causes an atypical-E2F-dependent delay of the G2/M transition, accompanied by reduced expression of E2F target genes involved in DNA repair. Live cell imaging of DNA damage revealed that cyclin F-dependent regulation of atypical E2Fs is critical for efficient DNA repair and cell cycle progression.**

**Keywords** cell cycle; DNA damage; E2F; proteolysis; SCF$^{cyclin\ F}$

**Subject Categories** Cell Cycle; DNA Replication & Repair; Post-translational Modifications & Proteolysis

**The EMBO Journal (2019) 38: e101430**

See also: **HI Rösner & CS Sørensen** (October 2019)

## Introduction

The atypical E2Fs, E2F7 and E2F8, are transcriptional repressors controlling a network of genes that drive cell cycle progression. Our previous studies have revealed that classical E2F7/8 target genes, such as *CDT1, CDC6,* and *RAD51,* are involved in DNA replication, repair, and metabolism (Westendorp *et al*, 2012; Kent *et al*, 2016). Ectopic expression of atypical E2Fs leads to downregulation of these target genes accompanied by a permanent S-phase arrest and severe DNA damage (Westendorp *et al*, 2012; Yuan *et al*, 2018). In contrast, depletion of E2F7 and E2F8 leads to upregulation of E2F targets, loss of DNA damage checkpoint control, and spontaneous development of hepatocellular carcinomas (Kent *et al*, 2016; Thurlings *et al*, 2016). As such, activity of E2F7 and E2F8 must be tightly regulated during the cell cycle and in response to DNA damage. Nonetheless, the regulation of the atypical E2Fs is not fully elucidated. Recently, we have shown that APC/C$^{Cdh1}$ targets E2F7 and E2F8 for degradation during the G1 phase of the cell cycle and that inhibition of the APC/C$^{Cdh1}$-mediated degradation of E2F7 and E2F8 impairs S-phase entry, eventually resulting in cell death (Boekhout *et al*, 2016). Additionally, in response to replication stress the repressor activity of atypical E2Fs is inhibited by checkpoint kinase 1 (Chk1) to prevent a permanent cell cycle arrest (Yuan *et al*, 2018). These studies demonstrated that the proper regulation of atypical E2Fs during cell cycle progression and DNA damage is critical to avoid a detrimental effect on cell survival.

In previous studies, while investigating the oscillating expression pattern of atypical E2Fs, we observed that the protein levels of E2F7 and E2F8 peak in S phase and are downregulated during the G2 phase of the cell cycle. However, the regulatory mechanism behind the downregulation in G2 is unknown (Boekhout *et al*, 2016). Since the transcript levels of E2F7 and E2F8 are only slightly lower in G2 compared to S phase, it is likely that atypical E2Fs are subjected to proteasomal degradation during this phase of the cell cycle. Previous studies have linked the E2F family members with G2-to-M transition (Ishida *et al*, 2001; Polager *et al*, 2002; Zhu *et al*, 2004). In addition to genes that are involved in DNA replication and repair, a substantial number of mitotic genes such as *CDK1, CCNB1,* and *PLK1* were also identified as E2F-regulated genes. We consistently found that E2F7 and E2F8 transcriptionally regulate a subset of genes that are related to chromatin and cytoskeleton organization

1 Department of Pathobiology, Faculty of Veterinary Medicine, Utrecht University, Utrecht, The Netherlands
2 Hubrecht Institute-KNAW and University Medical Center Utrecht, Utrecht, The Netherlands
3 Department of Biotechnology, University of Verona, Verona, Italy
4 Medical Microbiology, University Medical Center Utrecht, Utrecht, The Netherlands
5 Division Molecular Genetics, Department Pediatrics, University Medical Center Groningen, Groningen, The Netherlands
  *Corresponding author. Tel: +31 3025 35252; E-mail: b.westendorp@uu.nl
  **Corresponding author. Tel: +31 3025 34293; E-mail: a.debruin@uu.nl
  † These authors contributed equally to this work

(Westendorp *et al*, 2012). Together, these studies give rise to the research questions of how atypical E2Fs are regulated and what their function during G2 phase is.

We thereby focused on the potential involvement of the Skp–Cullin–F-box protein-containing complex (SCF), an E3 ubiquitin ligase complex that controls the transition between G1/S and G2/M phases by targeting a number of key cell cycle regulators for proteasomal degradation (Nakayama & Nakayama, 2005). The substrate specificity of the SCF complex is determined by the F-box protein subunits. To date, over 70 human F-box proteins have been identified, and the founding member of the F-box family is cyclin F. Mouse embryonic fibroblasts (MEFs) lacking cyclin F exhibited cell cycle defects, indicating that cyclin F plays a role in cell cycle regulation (Tetzlaff *et al*, 2004). In addition, emerging evidence supports the importance of cyclin F in promoting the G2/M phase transition and preventing genomic instability (D'Angiolella *et al*, 2010; Choudhury *et al*, 2016).

In the current study, we discovered that SCF^cyclin F targets E2F7 and E2F8 for proteasomal degradation during G2 phase in human cells. Inhibition of cyclin F-dependent E2F7/8 degradation caused a defect in G2 progression and increased DNA damage accompanied by downregulation of E2F target genes involved in DNA replication and DNA repair. These findings suggest that degradation of atypical E2Fs via cyclin F might be necessary for efficient repair of DNA lesions during G2. Taken together, this study provides new mechanistic insights into how human cells control the progression through G2 phase of the cell cycle.

# Results

### E2F7 and E2F8 are subjected to Cullin-RING ligase-dependent degradation during G2 and early M phase

Our previous study showed that E2F7 and E2F8 are substrates of APC/C^Cdh1 during G1 phase and that their protein levels peak during S phase when APC/C^Cdh1 is inactive (Boekhout *et al*, 2016). However, protein levels of atypical E2Fs already begin to decline during G2 phase, when APC/C^Cdh1 is still inactive. This suggests an additional mechanism targeting E2F7 and E2F8 proteins for degradation in G2. To monitor the protein levels of E2F7 and E2F8 throughout the cell cycle, HeLa cells were synchronized at the onset of S phase by double thymidine treatment and subsequently released into fresh medium. Both atypical E2Fs were already expressed at the onset of the double thymidine release, and their protein levels peaked 6 h after the release during late S phase (Figs 1A and EV1A). Notably, the levels of E2F7 and to a lesser extent E2F8 decreased 9–12 h after release when most cells were in G2 phase. Release from a hydroxyurea (HU) block also showed that E2F7/8 markedly decreased after 8 h when most cells were in G2 (Fig EV1B). In line with this, E2F7/8 protein levels were low in cells treated with nocodazole, a microtubule inhibitor that arrests cells in prophase (Fig 1B). Together these findings suggest that E2F7/8 peak in S phase and are degraded during G2 and early mitosis.

We investigated which mechanism could be responsible for degradation of E2F7/8 during G2 and prophase, and we reasoned that the SCF (Skp–Cullin–F-box protein) ubiquitin ligase complex would be a highly likely candidate (Nakayama & Nakayama, 2006).

The SCF is the largest member of E3 ligase family, and among its many functions is the control of G2/M phase transition by proteasomal degradation of key cell cycle regulators, including the APC/C^Cdh1 inhibitor Emi1 (Guardavaccaro *et al*, 2003; Margottin-Goguet *et al*, 2003; Herrero-Ruiz *et al*, 2014). We therefore tested whether the Cullin-RING ligase promotes the degradation of E2F7 and E2F8 by treating HeLa cells for 16 h with MLN4924, a potent and selective Cullin-RING ligases inhibitor (Soucy *et al*, 2009). To avoid bias from effects of this inhibitor on cell cycle progression, Hela cells were arrested in prophase with nocodazole. Under these conditions, the degradation of the atypical E2Fs was rescued by MLN4924, suggesting that E2F7/8 are targets of the SCF complex (Fig 1C). To test whether Cullin-RING ligase inhibition increases the half-life of E2F7 and E2F8, cells were treated with cycloheximide (CHX), to inhibit protein synthesis, in the presence or absence of MLN4924. Indeed, both E2F7 and E2F8 were stabilized by MLN4924 treatment (Fig 1D). These data demonstrate that atypical E2Fs are subjected to degradation by the Cullin-RING ligases during G2 and early M phase of the cell cycle.

### Cyclin F binds to E2F7 and E2F8 via defined C-terminal motifs

The SCF complex selectively binds to its substrates via specific F-box protein subunits (Nakayama & Nakayama, 2005). Since the degradation of E2F7/8 occurred during G2 and prophase, we therefore hypothesize that the F-box protein cyclin F, a SCF ubiquitin ligase complex that is also active in G2 phase, could be a putative candidate for E2F7/8 degradation. This atypical cyclin does not interact with cyclin-dependent kinases but instead functions as a conserved substrate recognition subunit of the SCF ubiquitin ligase complex. It mediates degradation of multiple proteins including SLBP, RRM2, and CDC6 during G2 phase, to control cell cycle progression and to maintain genome stability (D'Angiolella *et al*, 2012; Dankert *et al*, 2016; Walter *et al*, 2016). Previous work demonstrated that cyclin F can bind to its substrates via a cyclin-binding sequence (known as CY motif) which contains a hydrophobic patch RxL or RxI motifs (D'Angiolella *et al*, 2013). We mapped three conserved putative CY motifs within murine E2F7 and four within murine E2F8 (Fig 2A). Immunoprecipitation was performed to examine the interaction between cyclin F and E2F7/8. We overexpressed EGFP-tagged E2F7 and E2F8 or only EGFP and found that E2F7/8-EGFP, but not EGFP alone, interacts with endogenous cyclin F (Fig EV1C). Reciprocal immunoprecipitation showed that Flag-tagged cyclin F can also pull down exogenous E2F7/8-EGFP (Fig EV1D).

Next, we aimed to identify the cyclin F-binding motif in E2F7/8 and mutated RxL or RxI motifs to two alanines (AxA). A series of binding experiments using both wild-type and AxA mutants were carried out to evaluate their interactions with cyclin F (Fig 2B). We found that E2F7 and E2F8 with mutations at their C-terminal CY motifs (E2F7^RxL/AxA 894/896 and E2F8^RxL/AxA 860/862, hereafter abbreviated to E2F7^R894A and E2F8^R860A) failed to interact with endogenous cyclin F. These data provide strong evidence that cyclin F binds to both E2F7 and E2F8 via a canonical CY motif. E2F7 and E2F8 have highly similar amino acid sequences and these C-terminus motifs are located at parallel positions on E2F7 and E2F8. Furthermore, these C-terminal motifs are conserved across multiple species (Fig EV1E) suggesting that the interaction between cyclin F and atypical E2Fs also occurs in other species. We then performed

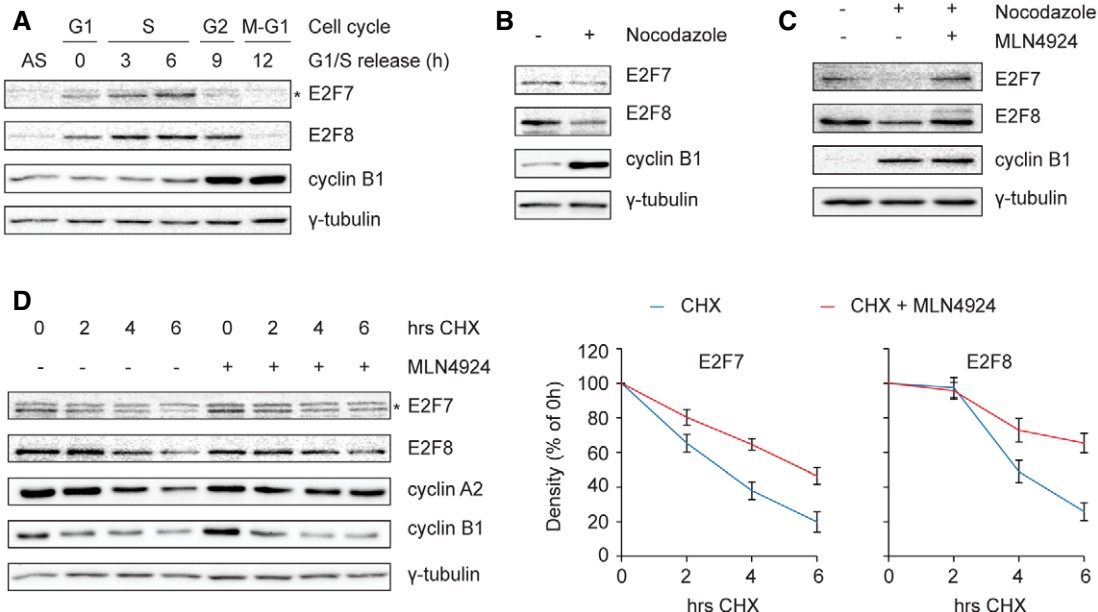

**Figure 1. E2F7 and E2F8 are subjected to Cullin-RING ligase-dependent degradation during G2 and early M phase.**

A  Protein levels of E2F7 and E2F8 during cell cycle progression. HeLa cells were synchronized by a double thymidine block and released into fresh medium. Cells were harvested at the indicated time points, and an asynchronous (AS) condition was used as control. Protein levels were measured by immunoblotting, and cell cycle progression was determined by flow cytometry (shown in Fig EV1A). The asterisk indicates the E2F7-specific band.

B  Decreased stability of E2F7 and E2F8 in nocodazole-arrested cells. HeLa cells were treated with either DMSO or nocodazole (50 ng/ml) for 16 h. Cells were harvested and lysed for immunoblotting. Protein expression of cyclin B1 was used as a marker for G2 or M, and γ-tubulin was used as loading control.

C  Selective Cullin-RING inhibitor MLN4924 rescued the degradation of E2F7/8 under nocodazole-arrested condition. HeLa cells were treated with DMSO, nocodazole, or nocodazole plus MLN4924 (0.1 μM) for 16 h. Cells were harvested and lysed for immunoblotting. Cyclin B1 expression was used as a marker for G2 or M cell cycle progression, and γ-tubulin was used as loading control.

D  Increased half-life of E2F7/8 by MLN4924 treatment. HeLa cells were treated with cycloheximide (CHX, 50 μg/ml) either with or without MLN4924 (0.1 μM). Protein levels of E2F7 and E2F8 were determined by immunoblotting (left panel). Asterisk indicates the E2F7-specific band. Cyclin B1 and cyclin A2 expressions were used as a marker for G2 or M cell cycle progression, and γ-tubulin was used as loading control. Quantifications (right panels) were performed based on two independent experiments. Bar and error bars represent mean ± SEM.

Source data are available online for this figure.

co-immunoprecipitations with truncated versions of cyclin F to test their interactions with EGFP-tagged E2F7/8. We found that a Δ270 mutant version of cyclin F, which lacks the cyclin domain, lost its binding to E2F7 while wild-type version and other truncated mutants still bound to E2F7 (Fig 2C). Of note, mutating the hydrophobic patch domain (ML/AA) of cyclin F did not interfere with its binding to E2F7. This suggests that the interaction between cyclin F and E2F7 required the cyclin domain, but not specifically via hydrophobic patch domain of cyclin F. EGFP-E2F8 interacted with all truncated or mutants versions of cyclin F, indicating that cyclin F binds via its F-box to E2F8 (Fig 2C).

## E2F7 and E2F8 are targeted for ubiquitination and degradation by cyclin F during G2/M phases

Since cyclin F interacts with atypical E2Fs, we hypothesized that over expression of cyclin F would result in de-stabilization of wild-type E2F7/8 but not of the E2F7$^{R894A}$ and E2F8$^{R860A}$ mutants that show reduced interaction with cyclin F. By performing co-transfection and immunoblotting, we did indeed observe that protein levels of wild-type E2F7 but not the E2F7$^{R894A}$ mutant were downregulated by over expression of cyclin F in G2/M phases, suggesting that

cyclin F mediates degradation of E2F7 via the motif at the C-terminus (Fig 3A). Overexpression of cyclin F decreased also the expression of endogenous E2F7/8 (Fig EV2A). The extent of down-regulation was similar to the effect of cyclin F overexpression on CDC6, a known cyclin F target (Walter *et al*, 2016). Interestingly, although the E2F8$^{R860A}$ mutant was more stabilized compared to wild-type E2F8 in G2/M phases, both versions were downregulated by cyclin F (Fig 3A). The E2F8$^{R408A}$ that showed also reduced inter-action with cyclin F (Fig 2B) was also degraded by cyclin F (Fig EV1F), suggesting that the degradation of E2F8 by cyclin F is not exclusively mediated through these conserved RxL interaction motifs.

If cyclin F targets E2F7 and E2F8 for degradation, then downregulation of cyclin F would result in stabilization of atypical E2Fs. To test this, cyclin F was knocked down by a pool of siRNAs and the protein expression of endogenous E2F7/8 was measured by immunoblotting. *Cyclin F* knockdown resulted in increased expression of E2F7/8 compared to cells transfected with a scrambled siRNA (Fig 3B). In line with this finding, we also showed that two different siRNAs against cyclin F lead to stabilization of endogenous E2F7/8 (Fig EV2B). In addition, we measured the half-life of E2F7/8 with CHX treatments and found that E2F7/8 were stabilized in the

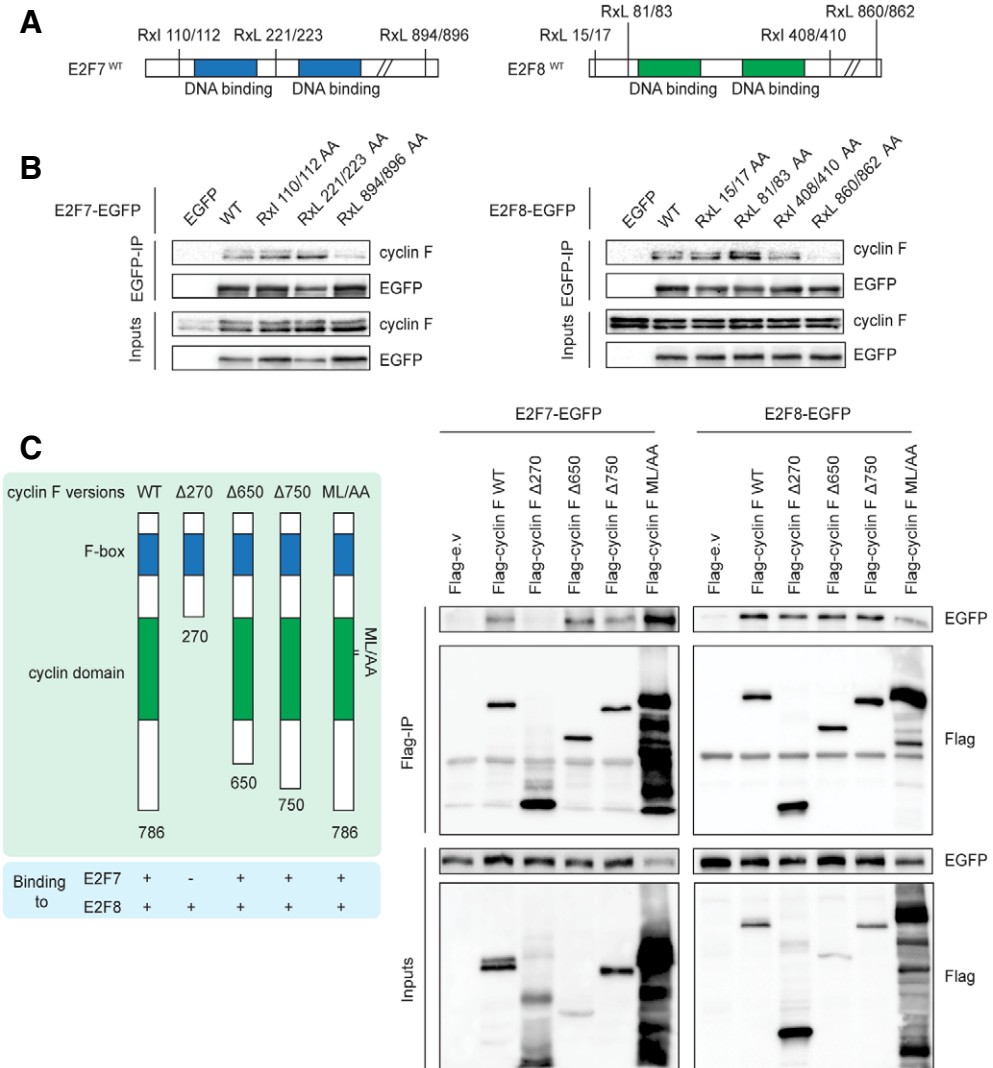

**Figure 2. Cyclin F binds to E2F7 and E2F8 through a defined motif at the C-terminus.**

A  Schematic view showing the location of putative cyclin F recognition motifs (RxL or RxI) on murine E2F7 and E2F8 proteins.

B  C-terminus motifs at parallel positions on E2F7 and E2F8 are essential for binding to cyclin F. The residues on each motif were mutated to alanines (R to A, I/L to A) with site-directed mutagenesis PCR. HEK293 cells were transfected with either EGFP-tagged empty vector (EGFP), wild-types E2F7/8 (WT), or alanine mutants. Nocodazole (50 ng/ml) was added 32 h after transfection, and MG132 (1 μg/ml) was added 5 h before harvesting at 48 h post-transfection. Cells were harvested and lysed for immunoprecipitation using anti-EGFP resin followed by immunoblotting with antibodies against cyclin F and EGFP.

C  Schematic view showing the truncated mutants and ML/AA mutant of cyclin F (left). HEK cells were transfected with the indicated constructs, and co-immunoprecipitation was performed using Flag resin (right).

Source data are available online for this figure.

presence of *cyclin F* siRNA compared to the scrambled siRNA (Fig EV2C). These data demonstrate that cyclin F targets E2F7/8 for degradation. To determine during which phase in the cell cycle this process occurs, we monitored the expression of atypical E2Fs during cell cycle progression after release from a double thymidine block in the presence and absence of *cyclin F* siRNA. We observed that protein levels of cyclin F gradually increased from early S phase and peaked 9 h after release, when most cells were in G2 phase (Figs 3C and EV2D). E2F7 levels started to decrease at that same time point. E2F8 proteins decreased later (12 h). At 12 h, when the majority of cells were still in G2, E2F7 and E2F8 protein

and transcript levels had almost completely disappeared (Figs 3C and EV2D and E). Importantly, *cyclin F* knockdown enhanced the protein levels of E2F7 and E2F8 at 9 h after thymidine release, when cells were in G2 phase. The mRNA levels of E2F7 were not affected by cyclin F knockdown (Fig EV2E), supporting that the stabilization of E2F7 resulted from reduced proteasomal degradation. E2F8 transcript levels were slightly higher at 0 and 9 h and lower at 3 and 6 h in *cyclin F* knockdown conditions compared to scr-treated cells. This finding suggests that increased transcript levels of E2F8 at 9 h might have contributed to the increased protein expression of E2F8.

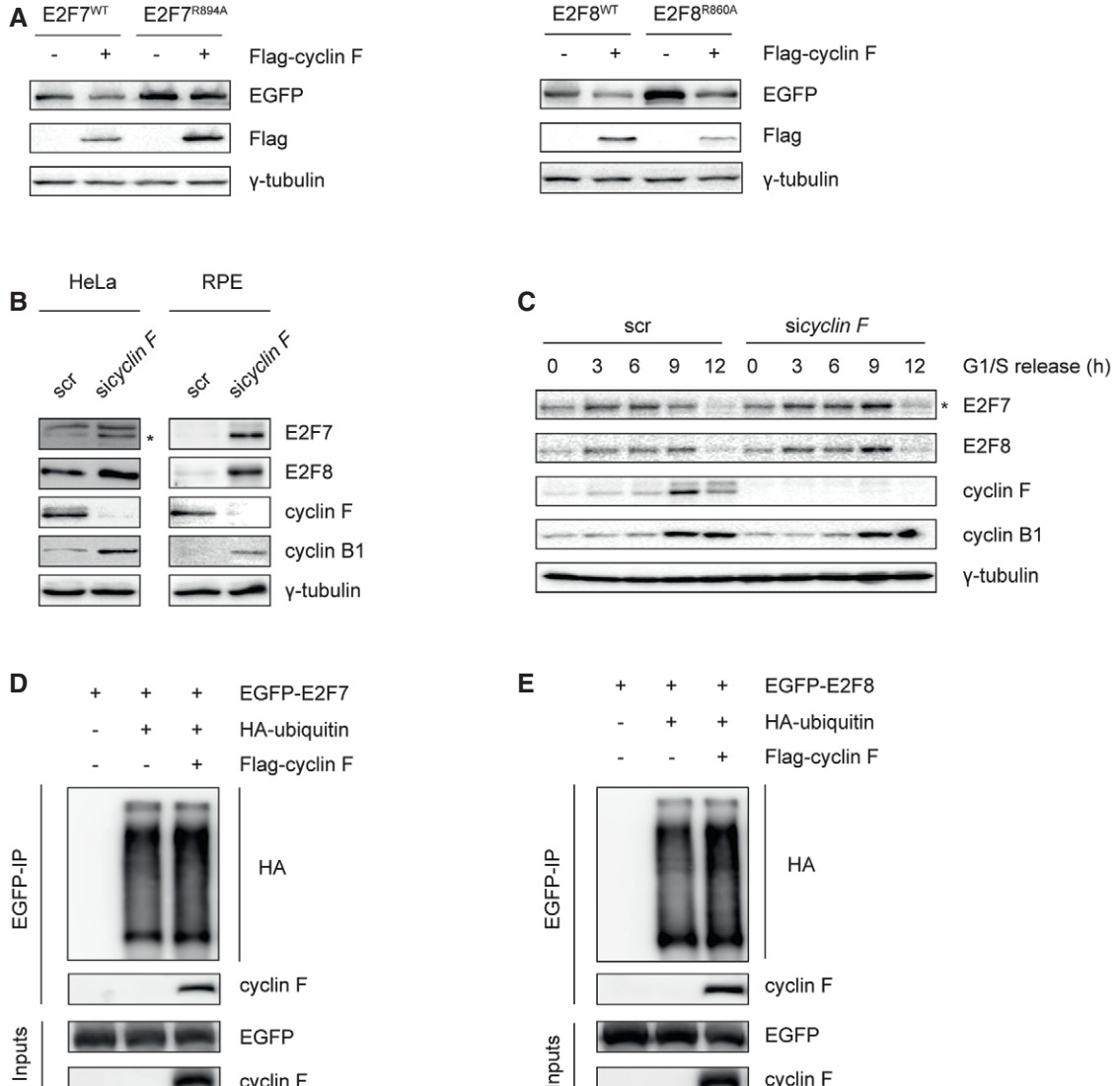

**Figure 3. E2F7 and E2F8 are targeted for ubiquitination and degradation by cyclin F during G2/M.**

A    Wild-type or mutant versions of EGFP-tagged E2F7/8 were co-transfected with either empty vector or Flag-tagged cyclin F in HEK293 cells. Nocodazole was added to cells 8 h before harvest. Forty-eight hours after transfection, cells were collected and lysed for immunoblotting.

B    Knockdown of cyclin F stabilized E2F7 and E2F8. HeLa and RPE cells were transfected with either scramble siRNA or pool cyclin F siRNA. Cells were harvested at 48 h post-transfection. Protein levels of E2F7/8 were analyzed by immunoblotting. Asterisk indicates the specific band of E2F7 detection.

C    Cyclin F targets atypical E2Fs during G2/M. HeLa cells were transfected with either scrambled siRNA (scr) or cyclin F siRNA (*sicyclin F*) for 24 h. Then, cells were synchronized by double thymidine block and released into fresh medium after the second block. Cells were harvested at the indicated time points after the release. Protein expression was measured by immunoblotting, and cell cycle progression was determined by flow cytometry (shown in Fig EV1A). Asterisk indicates the specific detection of endogenous E2F7.

D, E  Cyclin F contributes to the ubiquitination of E2F7 and E2F8 *in vivo*. HEK293 cells were transfected with HA-E2F7/8, with or without Flag-cyclin F, and with HA-tagged ubiquitin. Five hours before harvest, cells were treated with MG132. Forty-eight hours after transfection, HEK cells were harvested and lysed for immunoprecipitation pull-down assay with anti-HA resin followed by immunoblotting.

Source data are available online for this figure.

To verify whether cyclin F controls the stability of E2F7/8 through ubiquitin-mediated degradation, we performed *in vivo* ubiquitination assays. Atypical E2Fs and HA-tagged wild-type ubiquitin were co-expressed in the presence and absence of cyclin F. Then, E2F7 and E2F8 were subjected to immunoprecipitation followed by immunoblotting for HA-ubiquitin (Fig 3D and E). We found that E2F7 and E2F8 were poly-ubiquitinated. Over expression of cyclin F enhanced the ubiquitination of E2F7/8. In addition, we demonstrated that E2F7^R894A displayed a reduction in ubiquitination compared to E2F7^WT (Fig EV2F). Taken together, our data suggest that E2F7/8 are targeted for degradation by SCF^cyclin F-mediated ubiquitination.

## Failure to degrade E2F7 and E2F8 results in defected G2/M transition

Next, we aimed to investigate the biological significance of the cyclin F-dependent degradation of atypical E2Fs. In the flow cytometry data from Fig EV2D, knockdown of *cyclin F* induced a delay in the progression of cells through G2 or M phase, reflected by a smaller G1 cell population at 9 and 12 h after thymidine release. Given that protein levels of E2F7/8 were stabilized during G2 phase upon cyclin F depletion (Fig 3C), we hypothesized that the G2/M transition delay by cyclin F loss resulted from stabilized expression of E2F7 and E2F8. To test our hypothesis, we analyzed whether loss of E2F7/8 would rescue the cell cycle delay caused by loss of cyclin F. To this end, *E2F7* and *E2F8* (*7/8^(KO)*) were deleted in non-transformed human cells—retina pigment epithelial cells (RPE-hTERT) expressing the Fluorescent Ubiquitin Cell Cycle Indicator (FUCCI system) using CRISPR–CAS9 technology (Sakaue-Sawano *et al*, 2008). RPE-hTERT cells carrying a Cas9 construct lacking a small guiding RNA (sgRNA) were used as control (Ctrl). Complete and permanent deletion of both *E2F7* and *E2F8* was confirmed by immunoblotting (Fig EV3A). To monitor the cell cycle progression through the G2 and M phases, these two cell lines were synchronized at the onset of S phase by HU treatment for 16 h. After release from HU, the progression of each individual cell was recorded by live cell imaging (Fig 4A). Around 50% of Ctrl cells reached mitosis within 24 h after HU release, while only 20% of cells with *cyclin F* siRNA progressed through mitosis (Fig 4B). Importantly, the delay in cell cycle progression induced by the knockdown of cyclin F was completely rescued in cells with deletion of E2F7/8.

Since HU treatment results in DNA damage and loss of E2F7/8 leads to an impaired DNA damage response (Koc *et al*, 2004; Zalmas *et al*, 2008; Aksoy *et al*, 2012; Thurlings *et al*, 2016), the impact of cyclin F-mediated degradation of E2F7/8 was investigated under unperturbed conditions. Similar to the setting in Fig 4A, both Ctrl and *7/8^(KO)* cells were transfected with either scrambled siRNA or siRNA against *cyclin F* and subjected to fluorescent live cell imaging. Four cell cycle stages, i.e., G1 phase, G1–S transition, late S to G2, and M, were analyzed based on the fluorescence signal (see Materials and Methods). For each condition, 50 cells were followed and their cell cycle progression starting from G1 phase was recorded (Fig 4C). We found that 70% of the Ctrl cells (35/50) completed mitosis during the observed time window, while *cyclin F* knockdown resulted in a delayed cell cycle progression with only 44% (22/50) of all cells finishing mitosis in the same time period (Fig 4D). In line with the HU-synchronized cells, deletion of *E2F7/8* could rescue the delayed cell cycle progression induced by *cyclin F* knockdown under unperturbed conditions, with 60% of scrambled siRNA *7/8^(KO)* cells completing mitosis within 24 h compared to 58% (29/50) of *cyclin F* siRNA *7/8^(KO)* cells. Moreover, the time from G1–S transition to mitosis for those cells that completed this process was measured (Fig 4E). We found that Ctrl cells with *cyclin F* siRNA moved from S-phase entry to completion of mitosis in an average time of approximately 18 h, compared to < 16 h in Ctrl cells incubated with scrambled siRNA. This delayed cell cycle progression phenotype was absent in the *7/8^(KO)* cell lines treated with cyclin F siRNA; they also needed < 16 h to complete mitosis from the moment of S-phase entry. We also quantified the fates of the whole cell population (50/each, at the last frame of the live imaging).

Strikingly, 42% of the Ctrl cells with *cyclin F* knockdown were in late S or G2, compared to only 18% in scrambled condition (Fig 4F; individual cells are shown in Fig EV3B). More importantly, such delay was not observed in the *7/8^(KO)* cells, suggesting that the delay in S and/or G2 progression by *cyclin F* knockdown is a consequence of stabilized E2F7/8.

## Overexpression of E2F7^(R894A) mutant delays G2–M progression

If cyclin F-dependent degradation of atypical E2Fs is important for G2/M progression, then a non-degradable version of an atypical E2F should slow down cell cycle progression. To test this, we first compared the appearance of E2F7^(WT) and E2F7^(R894A) proteins when adding doxycycline immediately after HU release (Fig 5A). Immunoblotting analysis revealed that 6–12 h after the addition of doxycycline, the protein levels of mutant version E2F7^(R894A) were increased compared to E2F7^(WT), whereas mRNA levels of E2F7^(R894A) were lower than those of E2F7^(WT) (Fig 5B). This finding excluded the possibility that the enhanced expression of E2F7^(R894A) was related to higher transcript levels. Then, we compared the G2/M progression between the HeLa cell lines in which either E2F7^(WT) or E2F7^(R894A) was induced by doxycycline after HU release (Fig 5C). In this live cell imaging assay, over expression of E2F7^(WT) caused a minor cell cycle delay toward mitosis (log-rank $P = 0.054$), while E2F7^(R894A) significantly reduced the number of cells finishing mitosis after 24 h (log-rank $P < 0.01$). Interestingly, we found the γ-H2AX level was significantly higher in cells expressing E2F7^(R894A) than E2F7^(WT), suggesting that expressing E2F7^(R894A) induced DNA damage and thereby delayed cell cycle progression (Fig EV3C). Together, these data demonstrated that expression of mutant version E2F7^(R894A) resulted in delayed G2–M progression.

## Cyclin F controls transcription of DNA repair genes via degradation of E2F7/8

To determine in an unbiased manner which transcripts are regulated by atypical E2Fs in a cyclin F-dependent manner, we performed RNA sequencing on nocodazole-synchronized cells treated with scrambled (scr), *cyclin F*, *E2F7/8*, or *cyclin F/E2F7/8 (triple)* siRNAs. We observed a substantial overlap between genes that were downregulated by *cyclin F* siRNA compared to scr, and genes that were upregulated in *cyclin F/E2F7/8* siRNAs compared to *cyclin F* siRNAs (Figs 6A and EV4A). Gene ontology analysis showed that these genes, which were downregulated genes after cyclin F knockdown and rescued by additional E2F7/8 knockdown, were strongly enriched for DNA repair and replication pathways (Figs 6B and EV4B, Dataset EV1). Among these DNA repair genes, we observed many known E2F7/8 target genes, such as *RAD51*, *MSH2/6*, *EXO1*, and *CHEK1* (Westendorp *et al*, 2012). Quantitative PCR and immunoblotting on a subset of these DNA repair genes confirmed that they were indeed downregulated by cyclin F depletion in an E2F7/8-dependent manner (Fig 6C and D). Consistently, the expressions of E2F7/8 target genes involved in DNA replication showed a similar expression pattern (Fig 6C, lower panel). We also confirmed this finding in RPE cells (Fig EV4C). Interestingly, we found that genes known to control mitotic entry, such as *PLK1* (polo-like kinase 1) and *CCNB1* (cyclin B1), were upregulated in response to *cyclin F* knockdown (Fig EV4D), but were not affected by

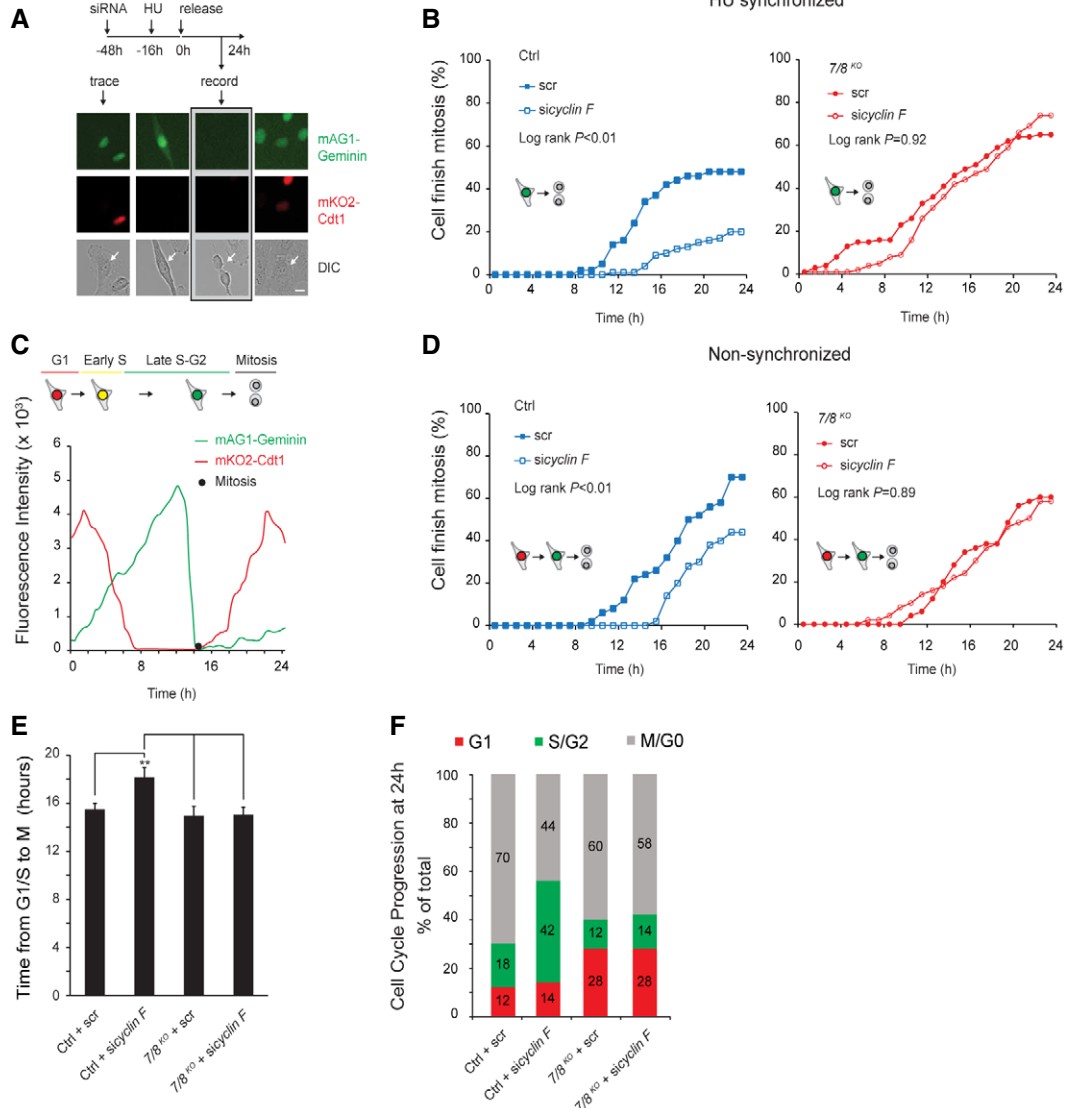

**Figure 4. Failure to degrade E2F7 and E2F8 results in delayed G2/M progression.**

A   Schematic view of the experimental setting for the HU-synchronized live cell imaging. Forty-eight hours before imaging, RPE-FUCCI cells were transfected with siRNA against scramble or cyclin F. Sixteen hours before imaging, cells were synchronized at the G1/S border by HU (2 mM) treatment. Representative images from different channels are shown, and white arrows in differential interference contrast (DIC) channel indicate the traced cell. Scale bar: 10 μm.

B   Quantification of the number of Ctrl (left panel) and *7/8^KO* (right panel) RPE-FUCCI cells with *scr* or *sicyclin F* that completed mitosis after HU release. For each condition, 100 cells were monitored by live cell imaging. Each cell was followed until it successfully finished mitosis and divided into two daughter cells for a maximum of 24 h. Log-rank tests were performed to analyze the statistical significance.

C   Schematic view of the experimental setting for live imaging of asynchronous cells. Forty-eight hours before imaging, RPE-FUCCI cells were transfected with siRNA against scramble or cyclin F. At the start of the imaging, G1 cells (red: mKO2-Cdt1 > mAG1-Geminin, 50 cells per condition) were enrolled and subsequently monitored though their entire cell cycle until mitosis. The black dot represents mitosis.

D   Knockdown of cyclin F causes a delay in mitotic entry that is dependent on E2F7 and E2F8. The number of Ctrl (left panel) or *7/8^KO* (right panel) RPE-FUCCI cells with *scr* or *cyclin F* RNAi that finished mitosis during live cell imaging is shown. For each condition, 50 cells at G1 were monitored by live cell imaging. Each cell was followed until it successfully progressed through S and G2 phases, finished mitosis, and divided into two daughter cells, for a maximum of 24 h. Log-rank tests were performed to analyze the statistical significance.

E   Loss of cyclin F delays the progression from G1/S transition to mitosis. Histogram shows the time from G1/S (mAG1-Geminin intensity increases to higher than 10% of the maximum value in three consecutive imaging frames) to completed mitosis. Only cells that finished mitosis were enrolled in this quantification. Student's *t*-test was used to test the statistical significance, and asterisks indicate the *P* < 0.01. Bar and error bars represent mean ± SEM.

F   Depletion of cyclin F stalls the cell cycle at late S/G2. After 24 h of live cell imaging, the cell cycle progression from panel (E) was quantified. Histogram shows the percentage of cells at each stage (at 24 h) over the whole population (50 cells per condition).

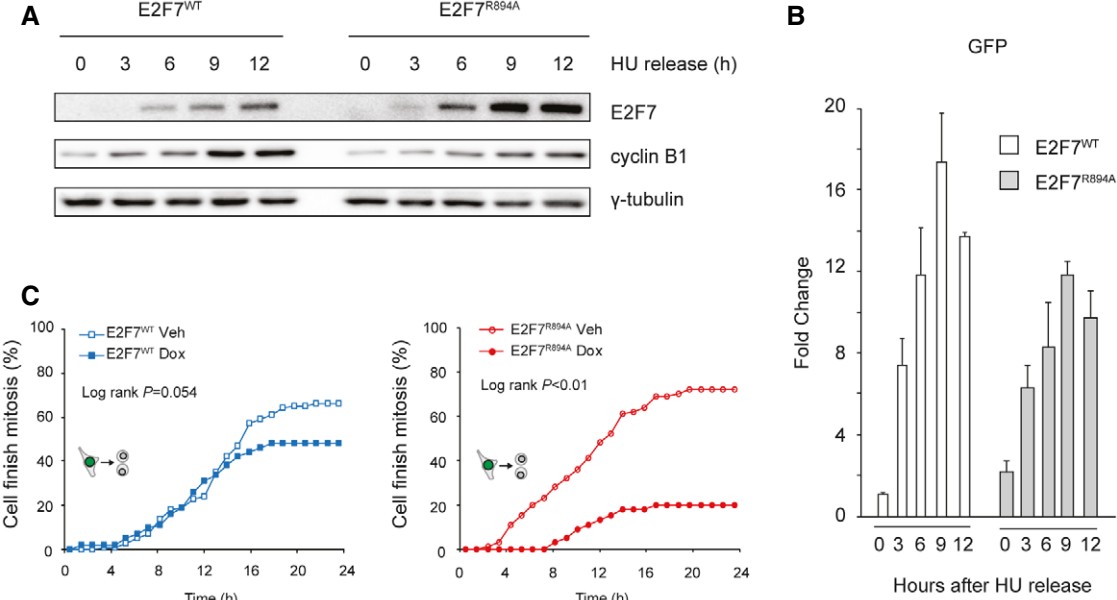

**Figure 5. Overexpression of E2F7R894A mutant delays G2/M progression.**

A Disruption of cyclin F binding site increased stability of E2F7. E2F7WT and E2F7R894A constructs were integrated in to HeLa/TO system. Cells were arrested with HU for 16 h before releasing into doxycycline containing medium, and cells were collected every 3 h for immunoblotting.

B qPCR showing that mRNA level of E2F7WT and E2F7R894A was at comparable levels. Bar and error bars represent mean ± SEM.

C Over expression of E2F7R894A delays cell cycle progression through G2–M phase. HeLa/TO cells expressing either wild-type or mutant version of E2F7 were arrested with HU for 16 h, and then, cells were released into fresh medium with or without doxycycline. Live cell imaging was performed to trace the G2–M progression of HeLa/TO cells.

Source data are available online for this figure.

knockdown of atypical E2Fs. This is in line with a previous study where it has been shown that cyclin F suppresses a B-Myb-driven transcriptional program regulating mitotic gene expression (Klein *et al*, 2015). Phosphorylated MPM2 as an indicator of M phase was slightly increased in si*cyclin F* and siTriple conditions compared to scr and si*E2F7+8* conditions, suggesting that depletion of cyclin F also affects mitotic entry through regulation of B-Myb target gene expression (Fig EV4E).

**Degradation of atypical E2Fs sustains DNA repair functions in G2**

Multiple DNA damage repair pathways, including mismatch repair (MMR), nucleotide excision repair (NER), base excision repair (BER), and homologous recombination (HR), are regulated by atypical E2Fs in a cyclin F-dependent manner (Fig 6B). We wondered if these transcriptional effects of cyclin F depletion would have functional consequences. Therefore, we tested if HR repair was impaired in cyclin F-depleted cells (Parvin *et al*, 2011). We found that knockdown of cyclin F significantly reduced HR repair efficiency, and this repair deficiency was fully recovered by additional knockdown E2F7 and E2F8 (Figs 7A and EV5A; see Materials and Methods).

If failure to degrade atypical E2Fs resulted in enhanced repression of DNA damage repair genes, then DNA lesions would accumulate. To test this idea, the level of phosphorylated γ-H2AX was measured using immunofluorescence staining in nocodazole-arrested cells (Fig 7B). Indeed, loss of *cyclin F* resulted in a significant increase of γ-H2AX levels, when compared with the

scrambled condition. In addition, combined knockdown of *E2F7/8* and *cyclin F* rescued the DNA lesions, suggesting that DNA repair capacity during G2 was restored by sustaining E2F-dependent DNA repair gene expression.

To monitor the dynamics of DNA damage repair by live cell imaging, a truncated version of 53BP1-mApple construct was integrated into RPE cells (Yang *et al*, 2015). In response to DNA damage, mApple-tagged 53BP1 localizes to damage sites, which can be seen as bright foci. Therefore, measurement of the numbers of 53BP1 foci in the nucleus can be used to monitor the onset and repair of DNA damage. RPE cells stably expressing this construct were transfected with siRNA targeting *cyclin F* or *E2F7/8* and then treated with HU for 16 h to arrest the cell cycle at the onset of S phase before live cell imaging. We first quantified the number of 53BP1 foci in the nucleus at the start of HU release. Interestingly, in the non-treated conditions, knockdown of *cyclin F* significantly increased the number of 53BP1 foci compared to scrambled siRNA (Fig EV5B). Furthermore, combined knockdown of *cyclin F* and *E2F7/8* attenuated this increase, suggesting that DNA damage repair function was restored. This result indicated again that the cell cycle delay caused by loss of *cyclin F* was due to a decrease in DNA repair capacity by enhanced repressor activity of E2F7/8. We noticed that the average number of 53BP1 foci per nucleus was significantly higher in scr and si*cyclin F* conditions compared to si*E2F7+8* and siTriple conditions when cells were synchronized with HU (Fig EV5B). This result suggested that depletion of E2F7 and E2F8 can elevate the DNA damage repair capacity not only to compensate

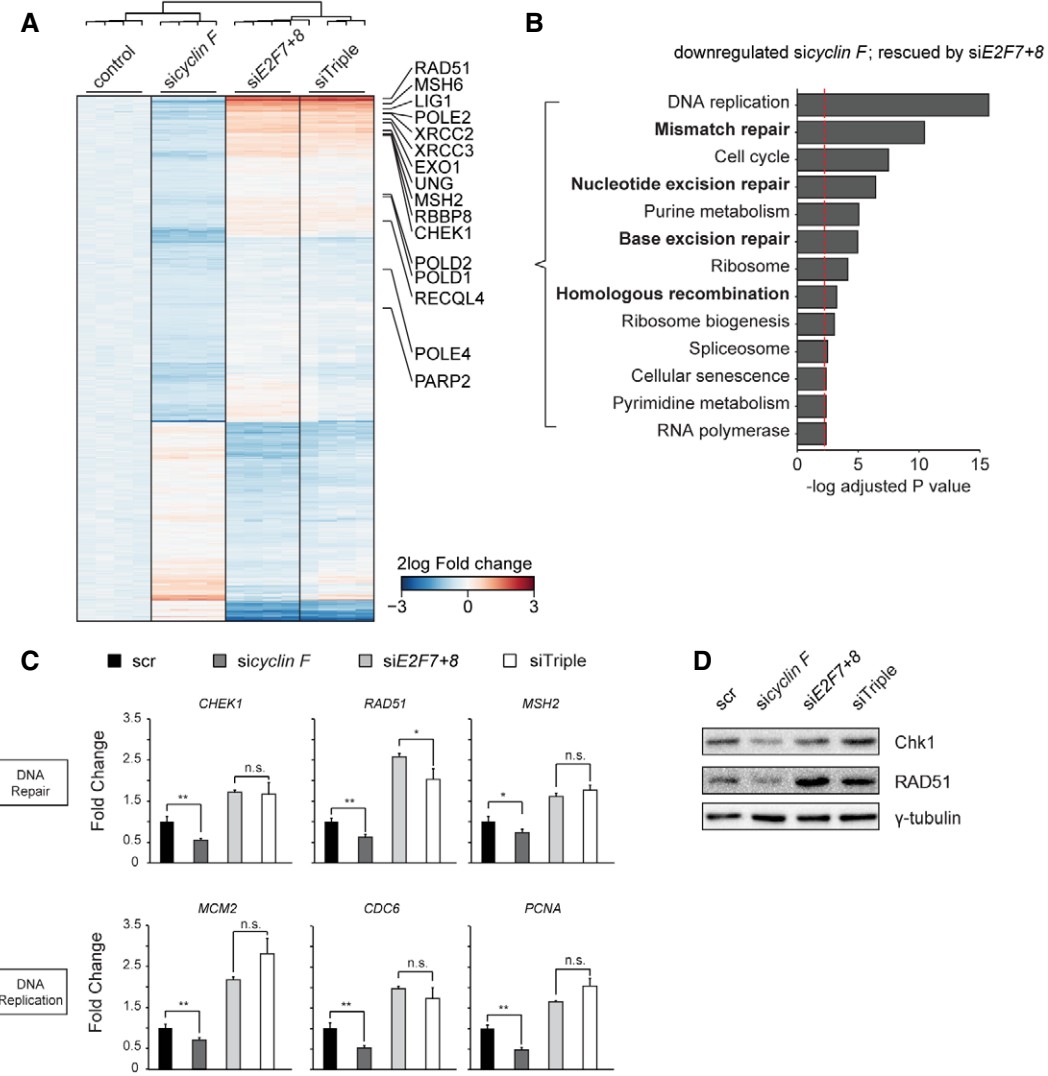

**Figure 6.  Cyclin F regulation of DNA replication and DNA repair genes is dependent on E2F7/8.**

A    Heatmap showing differentially expressed genes after cyclin F knockdown, and rescued by additional E2F7/8 depletion. Highlighted genes are all involved in DNA repair. HeLa cells were arrested with nocodazole for 16 h prior to harvesting to minimize bias from potential differences in cell cycle progression between the different conditions.

B    KEGG pathway analysis of genes downregulated by cyclin F knockdown, and rescued by additional E2F7/8 depletion. Bars represent −log P-values, such that larger values mean stronger statistical significance. The cutoff P-value 0.05 is shown as a red dotted line.

C    qPCR showing the RNA expression of atypical E2F target genes that are involved in DNA replication or DNA repair. HeLa cells were transfected for 48 h with siRNAs as indicated. Cells were incubated with nocodazole for 16 h before harvesting. Data represent averages ± SEM (n = 3); *P < 0.05 or **P < 0.01 (Student's t-test). n.s.: not significant.

D    Immunoblotting showing the protein levels of Chk1 and RAD51 in the indicated siRNA conditions. HeLa cells were treated with nocodazole for 16 h prior to harvesting.

Source data are available online for this figure.

the loss of cyclin F but also to an even higher level. This is consistent with our RNA-seq data showing that si*E2F7+8* and siTriple had enhanced expression of DNA repair genes compared to the control and si*cyclin F* conditions. We then quantified the DNA damage recovery time (from HU release until all 53BP1 foci disappeared) and cell division events of individual cells (Fig 7C and D). We found that loss of cyclin F significantly lengthened the damage recovery time to 16.4 h, compared to 11.1 h of scrambled condition. Furthermore, a substantial number of si*cyclin F* cells failed to recover from

DNA damage within 24 h, compared to scrambled control (Fig 7C). Most importantly, the DNA damage recovery time decreased to a length similar to the scrambled group by additional knockdown of *E2F7/8*, indicating that the delay of damage recovery was dependent on the atypical E2Fs functions. To further investigate when the DNA damage repair occurs after HU release, we measured the average 53BP1 foci number over time (Fig EV5C). We found that depletion of cyclin F caused a prolonged recovery from 53BP1 at 6–8 h after HU release. More importantly, this is the time window when the

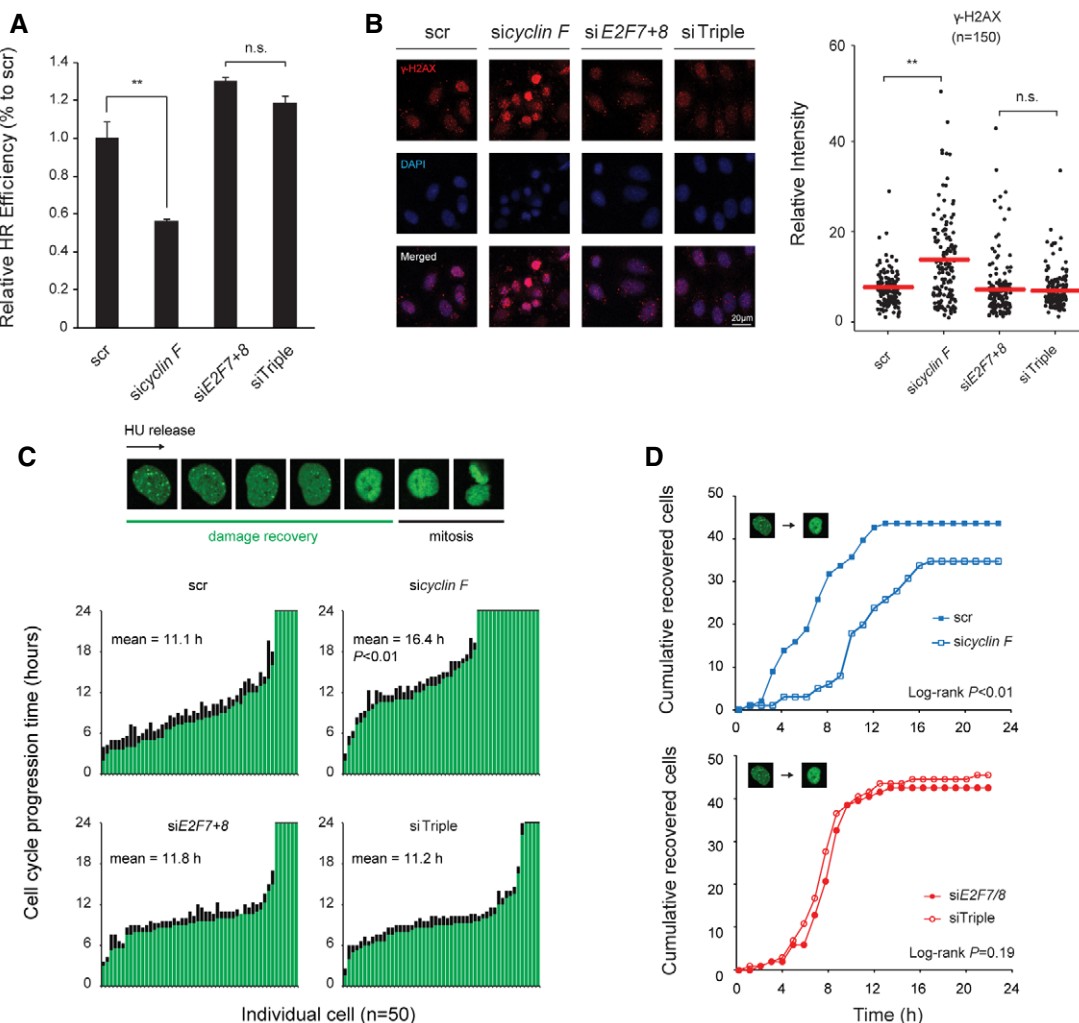

**Figure 7. Degradation of atypical E2Fs maintains DNA damage repair.**

A   Loss of *cyclin F* induced E2F7/8-dependent homologous recombination deficiency. HeLa cells that were stably transformed with pDR-GFP were transfected with siRNA as indicated. Forty-eight hours after the initial transfection, cells were harvested for flow cytometry. GFP-positive cells were gated (Fig EV5A). Relative HR efficiency was adjusted to the scramble siRNA condition. Data represent averages ± SEM (*n* = 3); **P < 0.01 (Student's *t*-test). n.s.: not significant.

B   Loss of *cyclin F* induced E2F7/8-dependent γ-H2AX accumulation. HeLa cells were transfected with the indicated siRNA for 24 h and then treated with nocodazole for 16 h before fixation for immunofluorescence staining of γ-H2AX. DAPI was used to stain the cell nucleus. Relative intensity of γ-H2AX was quantified by ImageJ software, and 150 cells were quantified for each condition. Red bars represent averages; **P < 0.01 (Student's *t*-test). n.s.: not significant. Scale bar 20 μm.

C   Loss of *cyclin F* increased DNA damage recovery time before cell division. RPE cells integrated with the 53BP1 construct were transfected with the indicated siRNA for 24 h and then treated with HU for 16 h. At the beginning of the imaging, only the single cells with at least one 53BP1 focus were traced, till the time frame that no 53BP1 foci were observed. The mitotic progression of the cells was defined as the duration from damage recovery to cell division. Histogram shows the damage recovery time (green) and the mitotic progression (black) of 50 cells for each condition. Chi-square analysis was performed to test the statistical significance (*P* < 0.01).

D   Knockdown of *cyclin F* caused a delay in DNA lesion recovery that is dependent on E2F7/8. The cumulative curves represent the add-up number of cells that overcome the DNA damage lesions, for a time frame of 24 h. For each condition, 50 cells were quantified. Log-rank tests were performed to analyze the statistical significance.

cells progressed to G2 phase (Fig EV5D), supporting our reasoning that cyclin F-dependent degradation of E2F7/8 impacts on G2 progression through regulation of DNA damage repair.

## Discussion

In the current study, we demonstrated a biological model that cyclin F-dependent degradation of atypical E2Fs is critical for DNA repair

and G2-phase progression (Fig 8). We first showed that E2F7 and E2F8 are targeted for degradation by cyclin F during G2/M phases. In an unperturbed cell cycle, cyclin F promotes the degradation of atypical E2Fs to allow a timely G2/M transition. Previous studies demonstrated that cyclin F functions as a key regulator of the cell cycle (Tetzlaff *et al*, 2004; Choudhury *et al*, 2016, 2017). *CCNF*, the gene encoding cyclin F, is highly conserved across different species. Moreover, its function is essential in the embryonic development of mice (Tetzlaff *et al*, 2004). MEFs (mouse embryonic fibroblasts)

derived from *Ccnf*$^{-/-}$ mice show reduced population doubling times and a delay in cell cycle re-entry from quiescence, indicating that cyclin F is required for cell proliferation. Interestingly, this slow-down in cell cycle re-entry may be partially explained by the inhibition of APC/C$^{Cdh1}$ by cyclin F during G0/G1 phase. Cdh1 is a substrate of cyclin F, and deletion of cyclin F resulted in stabilization of Cdh1 and inhibition of S-phase entry (Choudhury *et al*, 2016). However, it is important to note that cyclin F expression is low at G0/G1 and gradually increases after S phase, and most of cyclin F-mediated degradation occurs during G2 phase (D'Angiolella *et al*, 2012) (also shown in Fig 3C). These findings raised the question whether cyclin F regulates G2-phase progression and, if it does, by which mechanism. In this study, we showed that cyclin F knock-down leads to an E2F7/8-dependent G2/M transition delay. Most importantly, by using single live cell imaging, we demonstrated that this G2/M transition delay was likely due to a prolonged DNA repair period (Fig 7C). Cyclin F induces the degradation of E2F7/8 to maintain the expression of DNA repair genes, thereby ensuring flawless cell cycle progression through G2 until mitotic entry. Previous studies have shown that cyclin F can also regulate mitotic entry by suppressing B-Myb activity which promotes accumulation of crucial mitosis-promoting proteins (Klein *et al*, 2015). Our results are in line with this, because we observed that expression of key regulators of mitotic activity, such as *PLK1* and *CCNB1*, was

upregulated in cyclin F-depleted cells (Fig EV4D). These findings highlight two distinct roles of cyclin F during G2: regulation of DNA damage repair activity and preventing premature mitosis entry, through degradation of E2F7/8 and suppressing B-Myb, respectively.

Another central function of cyclin F is its role in guarding cells against genotoxic stress and genomic instability during the cell cycle. It has been shown that cyclin F promotes the degradation of the centrosomal protein CP110 and the DNA replication protein CDC6, thereby ensuring mitotic fidelity and preventing DNA re-replication (D'Angiolella *et al*, 2010; Walter *et al*, 2016). Moreover, cyclin F targets the ribonucleotide reductase RRM2 and stem loop binding protein SLBP for proteasomal degradation, which provides a balanced dNTP pool for DNA repair, and prevents SLBP-dependent accumulation of *H2AFX* mRNA translation to reduce susceptibility to genotoxic stress (D'Angiolella *et al*, 2012; Dankert *et al*, 2016). In line with these findings, our data demonstrate that cyclin F sustains the expression of DNA repair genes such as *RAD51*, *CHEK1*, and *BRCA1*, through degradation of the atypical E2Fs in G2 phase. Interestingly, in response to irradiation, cyclin F has been shown to be downregulated in an ATR-dependent manner, which resulted in stabilization of SLBP and RRM2 to promote DNA repair (D'Angiolella *et al*, 2012; Dankert *et al*, 2016). Interestingly, both *SLBP* and *RRM2* are *bona fide* targets of E2F7/8 (Westendorp *et al*, 2012; Kent *et al*, 2016). Therefore, we hypothesized that cyclin F controls SLBP and RRM2 expression at two different levels: directly via their ubiquitin-dependent degradation and indirectly via degradation of the transcriptional repressors E2F7/8. These complex regulation mechanisms mediated by cyclin F could be significant to the cancer field, since aberrant expression of RRM2 has been found in multiple types of cancers and failure to maintain the balance of dNTP can cause genome instability (Xu *et al*, 2008; Kumar *et al*, 2011; Ahluwalia & Schaaper, 2013).

In addition to *RRM2* and *SLBP*, cyclin F targets CDC6 for degradation, which is also transcriptionally regulated by atypical E2Fs (Westendorp *et al*, 2012; Kent *et al*, 2016). Thus, E2F-dependent transcription and SCF$^{cyclin F}$ appear to have partially overlapping functions. Therefore, the repressor functions of atypical E2Fs could potentially compensate for the loss of cyclin F. In addition, failure to degrade these overlapping targets (such as CDC6) in G2 phase could result in the re-initiation of DNA replication leading to genome instability (Walter *et al*, 2016). Therefore, we hypothesized that atypical E2Fs might act as a fail-safe mechanism to repress the expression of key cell cycle genes in case of inactivation of SCF$^{cyclin F}$. Such a compensation mechanism could help to minimize the occurrence of genome instability.

A recent study demonstrated that activator E2Fs (E2F1-3A) are also targeted by cyclin F for degradation during S and G2 phases (Clijsters *et al*, 2019). It raises a question whether the degradation of activator E2Fs and atypical E2Fs by cyclin F occurs simultaneously. Interestingly, a recent *in vivo* study demonstrated that there is a distinct difference in the timing of downregulating activator and atypical E2Fs (Cuitino *et al*, 2019). It was shown that E2F3A protein levels decrease in the middle of S phase, while E2F8 downregulation begins in late S phase. In addition, failure to degrade activators E2Fs or atypical E2Fs had distinct consequences on the cell cycle progression: E2F activator mutants that are unable to bind cyclin F induced premature S-phase entry, whereas non-degradable E2F7 mutants

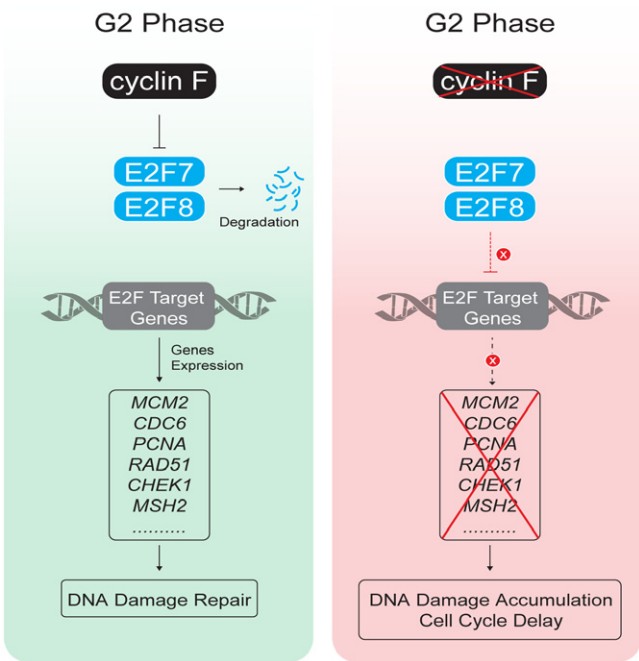

**Figure 8. Biological model of the current study.**

Schematic model of the current study. In an unperturbed G2 phase, cyclin F promotes the degradation of the transcription repressors E2F7/8, which leads to enhanced expression of their target genes, such as *RAD51*, *CHEK1*, and *MSH2* (left). Therefore, cyclin F-dependent degradation of E2F7/8 sustains the expression of DNA repair genes before mitosis. Inactivation of cyclin F results in stabilization of E2F7 and E2F8 in G2 phase (right). Active E2F7/8 at this stage repress the expression of DNA repair genes, leading to accumulation of DNA damage and delayed cell cycle progression.

delayed G2–M progression. These findings suggested that a proper cell cycle progression relies on a distinct regulation of activator E2Fs and repressor E2Fs by cyclin F. It is currently unclear how cyclin F can induce degradation of activator and atypical E2Fs at different time points of cell cycle, but it might depend on additional post-translational modification on E2Fs. Our data indicate that the biological significance to keep an intermediate level of E2F7/8 during the G2 phase is most likely to support DNA damage repair before a cell can enter mitosis. This is in line with our previous work, which showed that the DNA replication stress kinase Chk1 phosphorylates E2F7/8 to inhibit its transcriptional repressor function on DNA repair genes and thereby promotes DNA lesion recovery upon replication stress (Yuan *et al*, 2018). Moreover, two recent studies indicated that loss of E2F7 conferred resistance to DNA damaging drugs by elevating expression of DNA repair genes such as *RAD51, BRIP1,* and *FANCE* (Clements *et al*, 2018; Mitxelena *et al*, 2018). These results raise the question whether stabilization of E2F7/8 would in turn sensitize cancer cells toward chemotherapy.

To conclude, our study discovered a novel regulatory mechanism for atypical E2Fs whereby cyclin F mediates degradation during the G2 phase of the cell cycle. Degradation of E2F7 and E2F8 is of importance for proper G2 progression as depletion of cyclin F leads to a defect in cell cycle progression that depends on atypical E2Fs. Moreover, we provide novel insights into the regulation of DNA damage repair gene expression during G2/M phases, in which cyclin F-mediated degradation of atypical E2Fs promotes DNA damage repair by sustaining DNA repair gene transcription.

# Materials and Methods

### Cell culture, cell line generation, and transfection

HeLa, hTERT-RPE1, and HEK 293T cell lines were purchased from ATCC and cultured in DMEM (41966052; Thermo Fisher Scientific) containing 10% fetal bovine serum (10500064; Life Technologies). The HeLa cell line stably transformed with pDR-GFP was a gift from Prof. Dr. M.A.T.M. (Marcel) van Vugt, University of Groningen, The Netherlands. Site-directed mutagenesis was performed by a two-step PCR amplification (PCR protocol and primers are provided in Table EV1). Successful mutations were confirmed by Sanger sequencing (Macrogen, Inc.). Other drugs used in this study are as follows: Nocodazole (50 ng/ml, M1404; Sigma-Aldrich); Hydroxyurea (2 mM, H8627; Sigma-Aldrich); Thymidine (2 mM, T9250; Sigma-Aldrich); Cycloheximide (50 μg/ml, 01810; Sigma-Aldrich); MLN4924 (0.1 μM, MLN-4924; Active Biochem); and MG132 (1 μg/ml, Peptide International, IZL-3175-v_5mg).

To transfect HEK cells, 130 μg/ml PEI (Polyethylenimine, 23966; Polysciences) was mixed with the desired plasmids (15 μg) containing DMEM (ratio of 1:1). Mixtures were added directly to the cells and incubated for 6 h before being replaced with fresh media. ON-TARGETplus Smartpool siRNAs (2 nM) were products from GE Dharmacon; siRNA transfection was carried out according to the manufacturer's protocol using RNAiMAX (13778075; Thermo Fisher Scientific). The following siRNAs were used: Dharmacon L-003215-00-0005 (si*cyclin F*), Thermo Fisher HSS175354 Thermo Fisher HSS128758/HSS128760 (si*E2F8*), and Dharmacon D-001210-02-05 (Scrambled).

The lentiviral construct containing a truncated version of 53BP1 tagged with mApple was obtained from Addgene (Apple-53BP1trunc was a gift from Ralph Weissleder (Addgene plasmid # 69531; http://n2t.net/addgene:69531; RRID:Addgene_69531)). Lentivirus was produced by transfecting HEK 293T cells with 10ug lentiviral packaging plasmids (1:1:1) and 10 μg of the 53BP1 construct with PEI for 2 h. Then, 10 ml fresh medium was added and virus was harvested after 48 h. Three milliliters of virus containing medium and polybrene (8 μg/ml) was added to RPE cells for an incubation of 24 h. RPE cells containing the construct were selected with puromycin (1.0 μg/ml) for 5 days.

### Immunoprecipitation and immunoblotting

Cells were washed twice with cold PBS and collected by scraping and spinning. Cells were lysed in RIPA buffer composed of 50 mM Tris–HCl, 1 mM EDTA, 150 mM NaCl, 0.25% deoxycholic acid, 1% Nonidet P-40, 1 mM NaF and NaV$_3$O$_4$, and protease inhibitor cocktail (11873580001; Sigma-Aldrich) for 30 min on ice. Then, lysates were centrifuged at 12,000 *g* for 10 min to collect supernatants. Finally, Laemmli buffer was added, and the samples were subjected to SDS–PAGE and immunoblotting. For immunoprecipitations, cells were lysed in RIPA buffer and immunoprecipitations were carried out by incubating 20 μl of anti-FLAG M2 Affinity Gel (A2220; Sigma-Aldrich) or GFP-Trap (gta-20; Chromotek). After the pull-down, the agarose beads were washed three times with RIPA and PBS before proceeding to a standard SDS–PAGE and immunoblotting. All antibodies used in this paper are listed in Table EV1.

### Flow cytometry

Cells were harvested by trypsinization and subsequent fixation with 70% ethanol and overnight storage at 4°C. Before staining, cells were washed twice with ice-cold Tris-buffered saline (TBS) and resuspended with 500 μl staining buffer that contained 20 μg/ml propidium iodide (P4170; Sigma-Aldrich), 250 μg/ml RNase A (RNASEA-RO; ROCHE), and 0.1% bovine serum albumin (A8531; Sigma-Aldrich). Samples were loaded on a BD FACSCanto II flow cytometer. Cell cycle analysis was conducted using the Cell Cycle analysis function from FlowJo v10.0 software.

### RNA sequencing

Total mRNA was collected using Qiagen's RNeasy kits. Sequencing libraries were then prepared using the TruSeq Poly-A kit, according to the manufacturer's instructions. All 16 samples were pooled into one lane of an Illumina NextSeq 500 sequencer. Quality of the raw sequencing data was first checked using the program FastQC. Then, sequencing reads were trimmed for adapter sequences and mapped to the human genome (assembly hg38) using STAR version 2.4.2a. All mapped reads were counted using HTSeq version 0.6.1 in union mode. Library preparation, sequencing, and mapping were performed at the Utrecht Sequencing Facility (USEQ). The raw count data were then used to perform differential expression analysis using DESeq2. Heatmaps were created using the software package pheatmap, and represent fold changes calculated from normalized count data.

## Live imaging

For live cell imaging, 4,000 RPE-FUCCI cells were seeded into a glass-bottom μ-Slide 8-well plate. siRNA transfections were carried out the next day, and imaging started at 48 h after cell seeding. A Nikon Stochastic Optical Reconstruction Microscope (A1R-STORM) was used for live imaging. For each condition, 5 × 5 fields (63× magnification/field) were obtained. Auto-focus was set to capture photographs from GFP (488 nm), RFP/mApple (555 nm), and differential interference contrast (DIC) channels every 20 min for 24 h.

The software NIS-Elements version 4.51.01 was utilized for the quantification. For the HU arrest/release experiment, 100 cells were traced manually under the DIC channel as previously described (Yuan *et al*, 2018). For the asynchronized experiment, the quality of the movies was first improved with the auto-scale rolling balls option, with the radius set to 30. Each cell was marked and traced with the ROI (region of interest) function. In total, 50 individual cells were selected for each condition, and one additional blank ROI was made to rule out the background signal. The fluorescence intensity from 480- and 560-nm channels (based on each selected ROI) was obtained with the "Time measurement" option in ROI panel. Cell cycle stages were determined by the fluorescence signal intensities of CDT1-mKusabira Orange (mKO) and Geminin-mAzami Green (mAG): G1 stage: red, mKO signal > mAG signal; G1–S transition: yellow, mAG signal increases to 10% of maximum in three consequent frames; late S to G2: green, mAG signal > mKO signal; and M to early G1: colorless, disappearance of mAG signal, and evidence of mitotic division from differential interference contrast (DIC) image.

For quantification of 53BP1 foci in Fig EV5B, cell image was obtained from each time point and 50 cells per condition were randomly selected for foci counting.

### Quantitative PCR

Isolation of RNA, cDNA synthesis, and quantitative PCR were performed based on the manufacturers' instructions for QIAGEN (RNeasy Kits), Thermo Fisher Scientific (cDNA synthesis Kits), and Bio-Rad (SYBR Green Master Mix), respectively. Gene transcript levels were determined using the $\Delta\Delta C_t$ method for multiple-reference gene correction. β-Actin and GAPDH were used as references. qPCR primer sequences are provided in Table EV1.

### CRISPR–CAS9 knockout

RPE-*hTERT*-FUCCI cells (a kind gift from Prof. Rene Medema, Netherlands Cancer Institute) were transduced with a lentiviral expression vector encoding both Flag-tagged Cas9 and a single guide (sg) RNA sequence against E2F7 (sgRNA #1: GTGCTGCCAGCCCA GATATA, sgRNA #2: GAGCTAGAAACTTCTGGCAC) or E2F8 (sgRNA #1: GTTCCTCTGCCACTTCGTCA, sgRNA #2: GATCTCTGTT GCGGATCTCA) cloned into a pSicoR backbone as previously described (van Diemen *et al*, 2016). Lentiviral particles were produced by co-transfecting the pSicoR construct with third-generation packaging plasmids into 239T cells. The sgE2F7 and sgE2F8 vectors contained puromycin and blasticidin resistance cassettes, respectively, thus allowing for sequential selection of E2F7 and E2F8 mutant clones by manual picking. Indel mutations were confirmed with Sanger sequencing, and complete deletion of E2F7 and E2F8 was

verified by immunoblotting for E2F7/8. Cells expressing the vector containing only Cas9, but no sgRNA, served as control cell lines.

### Statistical analysis

Immunoblot, immunoprecipitation, flow cytometry, FACS sorting, and qPCR results were repeated three times unless otherwise described in the figure legends. Statistical analyses on qPCR were analyzed by Student's *t*-test. Statistical test on Fig 5C was analyzed by chi-square test, and cumulative curves from Figs 4B and C, and 5D were analyzed by log-rank tests.

## Data availability

The RNA-sequencing data have been deposited in NCBI's Gene Expression Omnibus (Polager *et al*, 2002) and are accessible through GEO Series accession number GSE133416 (https://www.ncbi.nlm.nih.gov/geo/query/acc.cgi?acc=GSE133416). Downregulated genes in *cyclin F* knockdown and rescued by additional E2F7/8 knockdown are shown in Dataset EV1.

**Expanded View** for this article is available online.

## Acknowledgements

We thank Richard Wubbolts and Esther van 't Veld in the Center for Cellular Imaging (Faculty of Veterinary Medicine, Utrecht University, NL) provided technical support with the live cell imaging. We thank Marcel van Vugt for kindly providing the stable pDR-GFP HeLa cell line and Michele Pagano for the truncated cyclin F constructs. This work is financially supported by the China Scholarship Council (CSC) (File No. 201306380101) to Ruixue Yuan and (File No. 201706140153) to Qingwu Liu; by KWF Kankerbestrijding (Dutch Cancer Society) funding (KWF: UU2013-5777) to Bart Westendorp and Alain de Bruin and (KWF: UU2012-5667) to Robert Jan Lebbink; by the Netherlands Organization for Scientific Research (NWO: ALW-IN11-28) to Alain de Bruin; and by a Dutch Cancer Society grant (HUBR 2014-6806) to Daniele Guardavaccaro.

## Author contributions

RY and QL performed the experiments. HAS helped quantification of the live cell imaging and generated *E2F7/8* mutant cell lines. LY performed the ubiquitination assays, DG kindly provided the plasmid constructs encoding F-box proteins and provided valuable input in the identification of cyclin F as E2F7/8 binding partner. RJL designed and created the lentiviral CRISPR constructs. RY, DG, BW, and AdB conceived the study, experimental approaches, and data analysis and wrote the manuscript.

## Conflict of interest

The authors declare that they have no conflict of interest.

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
