## [Review Process File · The EMBO Journal]

Cyclin F-dependent degradation of E2F7 is critical for DNA repair and G2-phase progression

Ruixue Yuan, Qingwu Liu, Hendrika A Segeren, Laurensia Yuniati, Daniele Guardavaccaro, Robert Jan Lebbink, Bart Westendorp and Alain de Bruin.

Review timeline:

Submission date:	24 th December 2018
Editorial Decision:	20 th February 2019
Revision received:	20 th July 2019
Editorial Decision:	31 st July 2019
Revision received:	10 th August 2019
Accepted:	13 th August 2019

Editor: Hartmut Vodermaier

Transaction Report:

1st Editorial Decision

20th February 2019

Thank you for submitting your manuscript for our editorial consideration, and please excuse our delayed response caused by a considerable submission backlog and limited referee availability during/after the end-of-the-year holiday period. We have now received a complete set of reports from three expert reviewers, which are copied below for your information. As you will see, all referees consider your study and its results interesting in principle. At the same time, they however raise a substantive number of major concerns related mostly to the strength and conclusiveness of the data in support of the key interpretations, which would have to be decisively addressed before further consideration for publication would be warranted. In particular, it would be important to add various essential controls, test and rule out various alternative scenarios/explanations, and to deepen certain aspects of the analysis, especially regarding the significance of repair factor alterations for effects on G2 progression.

Given the general interest expressed by the referees and the fact that all of them offer constructive suggestions for deepening the insights and improving the conclusiveness of this work, I would in this case be open to considering a revised manuscript. It is however apparent that adequately addressing all key criticisms may require substantial further time and efforts, and may also be of uncertain outcome - I hope you therefore appreciate that I am presently not able to provide definitive

commitments for eventual acceptance on this occasion.

REFEREE REPORTS

Referee #1:

Yuan et al.

Cyclin F-dependent degradation of atypical E2Fs is critical for DNA repair and G2-phase progression

Overview/Summary

The E2F family of transcriptional regulators play key roles in cell cycle progression. E2F7 and E2F8 have particularly interesting functions in that, unlike the canonical E2Fs (1-3) which promote cell cycle progression by activating the expression of key cell cycle genes, E2F7 and E2F8 are instead transcriptional repressors. Accordingly, E2F7 and E2F8 represent potential tumor suppressors. Previous studies from the de Bruin lab (and others) demonstrated that the abundance of E2F7 and E2F8 strongly oscillates during progression through the cell cycle. Notably, both are degraded in G1-phase by the APC/C, a multi-subunit E3 ubiquitin ligase and core component of the cell cycle oscillator. However, it was also noted that the levels of both E2F7 and 8 decrease prior to G1, in the preceding G2-phase of the cell cycle. However, it was mechanistically unknown how this degradation was controlled.

Yuan et al. now show that this degradation is likely mediated by the SCF-Cyclin F ubiquitin ligase. The SCF ligases are modular E3 ubiquitin ligases that use F-box proteins to bind substrates and recruit them to the ubiquitin machinery. Cyclin F is the founding member of the F-box protein family, and unlike other "Cyclins", Cyclin F neither binds nor activates a CDK. The authors provide compelling data that Cyclin F regulates E2F7, and possibly E2F8, degradation in G2. They also show that phenotypes associated with Cyclin F loss are overcome by the simultaneous loss of E2F7/8, suggesting that Cyclin F functions upstream of E2F7/8 in cell cycle. The data in this paper is compelling and would be of interest to the cell cycle field. Nevertheless, there are several key points that need to be addressed before publication of this study.

Major Points

1. Figures EV1C and EV1D are key experiments, showing that Cyclin F depletion increases E2F7 and E2F8 abundance. It is unclear why they are in the supplement. However, there are problems with this experiment in its current form. First, it must be done using multiple RNAi reagents, as it is no longer sufficient to show the key finding of the paper using a single RNAi reagent. This would be greatly aided by a rescue experiment (although not req'd). Second, they must also blot for cell cycle markers (not E2F7/8) since depletion of Cyclin F is going to mess up the cell cycle and also presumably cause some DNA damage. Thus, this experiment should be repeated in both cell lines, with multiple siRNA reagents, and additional cell cycle markers.
2. The authors need to perform a reciprocal IP to show that Cyclin F can co-IP E2F7/8.
3. In Figure 2A, a loading control for E2F7 and E2F8 is needed. Did they use an empty vector, as indicated in the figure, or an EGFP expressing vector, as indicated in the text?
4. In Figure 3B it is not at all evident that the R860A mutant prevents E2F8 degradation. Considering its levels are much higher to start, Cyclin F is more poorly expressed, and that it still decreases significantly, it appears that despite the weakened interaction, this might not be the relevant site. This calls into question whether E2F8 is a bona fide substrate.
5. The ubiquitination data for E2F7 is not that convincing.
6. The authors fail to fully characterize their E2F mutants, which would go a long way to making the

convincing argument that these are indeed substrates. They should either show that the mutants fail to be ubiquitinated, or similarly, that they indeed have a longer half-life.

7. Why are E2F7 and E2F8 being degraded at the 12hr time point in the cell cycle experiment shown in 3D? It appears the cells are still in G2/M.

8. The authors need to show in synchronized cells that E2F7/8 mRNA levels are not affected by Cyclin F depletion. This is a concern because 7/8 levels decrease at later time pts in the cell cycle experiment mentioned above, and also, because the mutant E2F8 still appears to be degraded.

9. The authors have generated a very useful allele for determining how E2F7 regulation contributes to cell cycle. The other experiments in Figures 4 and 5 are interesting, but only indicate what happens when Cyclin F is absent and all of its other (known and unknown) substrates go up. These experiments show epistasis, but don't point to a role for Cyclin F regulation of E2F7/8. Why didn't the authors express WT and non-degradable mutants and determine the effect on cell cycle.

10. Why in the last two figures do the authors switch between using two entirely different assay systems? It makes it less convincing that the phenotypes are broadly observable.

11. It is unclear that the damage phenotypes have to do with Cyclin F regulation of 7/8 is G2. DNA damage accumulates only during DNA replication. How do the authors know that the cells are G2 arrested during the recovery?

Minor Points

1. In figure 1, the authors show that E2F7/8 abundance is regulated by MLN4924, a small molecule that inhibits all known Cullin RING ligases (not just the SCF ligases, which are a subfamily of this much larger class). They need to clarify the language in the manuscript to reflect that the regulation of E2F7/8 by MLN4924 does not indicate SCF regulation, but instead regulation by this broader class.

2. Authors mention that EMI1 is regulated by SCF, but only cite one of the relevant papers. Another, from the Pagano lab, should also be cited.

3. Figure 3C should be somehow quantified since the degree of change appears minimal, especially since there is more E2F7 in the zero time pt.

4. It is unclear why the authors decided to focus on Cyclin F. This should be explained.

5. There is a typo in the text referring to a hydrophobic "Batch" (change to patch).

6. The authors cannot conclude from Figure 3B that the "degree" of E2F7/8 downregulation is similar to that of Cdc6. These are not quantitative assays or blots.

7. In Figure 1D a loading control for another cell cycle protein should be shown, since MLN4924 is known to arrest cells in S/G2 and cause re-replication.

Referee #2:

Cyclin F-dependent degradation of atypical E2Fs is critical for DNA repair and G2-phase progression

The Authors propose that in addition to a previously described degradation pathway via APC/C, the atypical E2Fs (E2F7 and E2F8) are ubiquitinated by SCF-Cyclin F. This advances their turnover especially in G2 phase of the cell cycle. Deficiency in this mechanism is then linked with enhanced E2F7/8 repressor activity, which in particular has a negative impact on genome maintenance. In terms of relevance and novelty, this manuscript has merit. Firstly, Cyclin F is an essential protein

with several emerging functions that remains to be fully understood. Secondly, though E2F7/8 are functionally better understood, their turnover late in G2 and early M phase was not well understood. In general, this manuscript is coherent and logical. However, considerable part of the data are not sufficiently comprehensive and they are at times also of low quality. In essence, several crucial data sets are absent in the current manuscript as outlined further below.

Major points:

1. The use of pooled siRNA needs to be validated for Cyclin F as one off-target siRNA could mediate the phenotype. Single siRNAs should be purchased and their efficacy documented. The key phenotypic analysis should be repeated with single siRNA depletion.
2. The authors present data regarding the interaction between Cyclin F and E2F7/8. However, the mapping must also include Cyclin F domains to back up the ubiquitin ligase claim. Is the interaction dependent on the Cyclin domain region?
3. The investigation of E2F7/8 protein levels and turnover should be carried out with significantly more scrutiny, as the presented data show a minor if any change (Figure 3). E2F8 is not a convincing target for Cyclin F. Further to Figure 3C: How can Cyclin F mediate turnover of E2F7/8 when it is absent (scr, 2h-4h-6h Chx)?
4. The transcript data are rather weak and very biased. RNA-seq data should be included (+/-Cyclin F; +/-E2F7/8).
5. Evidence for changes in protein levels is crucial for key factors (RAD51, and others identified in point 4).
6. Repair deficiency must be validated, are cell systems (those mentioned in above point 4.) HRR-deficient or are other repair pathways involved?

Additional points:

1. Abstract claim in last sentence "...essential for DNA repair and cell cycle progression" is an overstatement.
2. Inputs vary (Fig. 2C, 3A), there is a need to repeat and replace. Many of the immunoblot panels appear to be substantially image processed, this greatly impacts figure quality and must be addressed (Fig. 1A, 1C, 1D, 2A, 3B, 3D, 3E EV1B, EV2C). The authors also should take care to present data without nocodazole (could be supplemental).
3. Figure 4: using 16h HU for synchronizing might not be the method of choice for CCNF-KD. Aphidicolin is a better choice.
4. Figure 4B+D: Could rather be presented in %. (same scale, but easier to grasp)
5. Foci number cannot be presented with two decimals.
6. Text links to EV2E and EV2F needs correction.
7. Replication fork restart processes may contribute to phenotypes, this should be mentioned.

Referee #3:

In this manuscript the authors have investigated the regulation of the atypical E2Fs 7 and 8 during G2 phase. They find that E2F7 and 8 are targets of Cyclin F-mediated degradation and identify the Cyclin F binding domains. Further, they show that loss of E2F7/8 rescues cell cycle defects by Cyclin F degradation. They also find that multiple known Cyclin F targets, involved in DNA replication and repair, are affected by E2F7/8 loss and propose that E2F7/8 may function as an inherent compensation mechanism in case of reduced Cyclin F function.

In general, the manuscript is well written and the study is well performed. The identification of

E2F7/8 as Cyclin F targets has clear implications for genome stability and cell cycle control. However, I have three main objections that I think should be addressed.

First, quite some of the measurements are performed in Nocodazole arrested cells, with the reasoning that it is important to compare cells in similar cell cycle phases. While I completely agree to this reasoning, depending on cell type and concentration, Nocodazole typically blocks most cells in prometaphase (not prophase), but can also as a stress-response block cells in G2. As the various treatments (in Fig 5 and others) impact the G2/M transition, it is important to show the distribution of G2 vs M in Nocodazole treated samples after E2F7/8 and Cyclin F RNAi.

Second, large parts of the reasoning focus on that Cyclin F-E2F7/8 impact on G2 progression through regulation of DNA repair. Whereas I agree that the data supports this as a contributing factor, the relative importance versus repression of cell cycle promoting factors as Cdk1, Cyclin B or Plk1 is not clear. This should be discussed.

Third, 53BP1 binding to damaged DNA is counteracted by mitotic kinases - which are gradually activated through G2 phase - and the absence of 53BP1 foci in cells close to mitosis cannot be taken for a guarantee that damaged DNA is repaired. Line 303-305, it should be specified that these are cells containing DNA damage due to HU treatment. Line 281-282, to make this point a mitotic kinase-independent measurement of DNA damage is required.

Minor points:

Lane 319, I don't find this figure.

Figure 4F - it's unclear to me whether this looks at cells that did not complete mitosis (lane 228) or the whole population (lane 651). I think that the addition of a (EV) figure similar to 5C showing the durations of all cells in fig 4C-F would help the interpretation.

I'd suggest a visual schematic figure for the proposed role of Cyclin F and E2F7/8 on target gene expression.

It would be good if asterisks on blots were used consistently - for example in fig1 it indicates specific band, in Ev1 unspecific band, and in some E2F7 stainings with multiple bands no asterisk is included.

The reference list contains many doublets.

1st Revision - authors' response

20h July 2019

Referee #1:

Yuan et al.

Cyclin F-dependent degradation of atypical E2Fs is critical for DNA repair and G2-phase progression

Overview/Summary

The E2F family of transcriptional regulators play key roles in cell cycle progression. E2F7 and E2F8 have particularly interesting functions in that, unlike the canonical E2Fs (1-3) which promote cell cycle progression by activating the expression of key cell cycle genes, E2F7 and E2F8 are instead transcriptional repressors. Accordingly, E2F7 and E2F8 represent potential tumor suppressors. Previous studies from the de Bruin lab (and others) demonstrated that the abundance of E2F7 and E2F8 strongly oscillates during progression through the cell cycle. Notably, both are degraded in G1-phase by the APC/C, a multi-subunit E3 ubiquitin ligase and core component of the cell cycle oscillator. However, it was also noted that the levels of both E2F7 and 8 decrease prior to G1, in the preceding G2-phase of the cell cycle. However, it was mechanistically unknown how this degradation was controlled.

Yuan *et al.* now show that this degradation is likely mediated by the SCF-Cyclin F ubiquitin ligase. The SCF ligases are modular E3 ubiquitin ligases that use F-box proteins to bind substrates and recruit them to the ubiquitin machinery. Cyclin F is the founding member of the F-box protein family, and unlike other "Cyclins", Cyclin F neither binds nor activates a CDK. The authors provide compelling data that Cyclin F regulates E2F7, and possibly E2F8, degradation in G2. They also show that phenotypes associated with Cyclin F loss are overcome by the simultaneous loss of E2F7/8, suggesting that Cyclin F functions upstream of E2F7/8 in cell cycle. The data in this paper is compelling and would be of interest to the cell cycle field. Nevertheless, there are several key points that need to be addressed before publication of this study.

Major Points

1. Figures EV1C and EV1D are key experiments, showing that Cyclin F depletion increases E2F7 and E2F8 abundance. It is unclear why they are in the supplement. However, there are problems with this experiment in its current form. First, it must be done using multiple RNAi reagents, as it is no longer sufficient to show the key finding of the paper using a single RNAi reagent. This would be greatly aided by a rescue experiment (although not req'd). Second, they must also blot for cell cycle markers (not E2F7/8) since depletion of Cyclin F is going to mess up the cell cycle and also presumably cause some DNA damage. Thus, this experiment should be repeated in both cell lines, with multiple siRNA reagents, and additional cell cycle markers.

Response: Based on the reviewer suggestion we moved the western blots showing that cyclin F knockdown increases expression of E2F7/8 (original Fig EV1C) to the main figures (new Fig. 3B). The original Fig. EV1D showing the cell cycle profiles of scr and cyclin F siRNA treated cells was to verify that the cells were appropriately synchronized for the E2F7/8 expression studies in Fig. 3C. We therefore left these in the supplemental figures, now new Fig. EV2D.

Furthermore, we ordered single RNAi reagent targeting cyclin F, the data is shown in Fig EV2B. Consistently, in both HeLa and RPE cell lines, individual siRNA against cyclin F leads to stabilization of E2F7 and E2F8. Furthermore new RNA sequencing experiments were performed with single siRNA for cyclin F (Fig.6). We measured also the cyclin A2 and cyclin B1 as cell cycle markers. Protein levels of cyclin A2 levels are comparable among all conditions, while cyclin B1 levels slightly varied under *si*cyclin F conditions. These findings suggest that knockdown of cyclin F has no major effect on the expression of these cell cycle markers.

2. The authors need to perform a reciprocal IP to show that Cyclin F can co-IP E2F7/8

Response: We performed reciprocal IPs and the data are shown in Fig EV1C and D. Our results demonstrated that cyclin F can indeed pull down E2F7/8. We performed also new co-IP studies with truncated version of cyclin F to further characterize the interaction between cyclin F and atypical E2Fs (Fig 2C).

3. In Figure 2A, a loading control for E2F7 and E2F8 is needed. Did they use an empty vector, as indicated in the figure, or an EGFP expressing vector, as indicated in the text?

Response: We used an EGFP expressing vector for this experiment. We loaded the same set of input samples again and showed the EGFP expression, as well as the EGFP-tagged E2F7/8 expression in the input samples (Fig EV1C). To clarify the result, we changed the figure text from e.v to EGFP.

4. In Figure 3B it is not at all evident that the R860A mutant prevents E2F8 degradation. Considering its levels are much higher to start, Cyclin F is more poorly expressed, and that it still decreases significantly, it appears that despite the weakened interaction, this might not be the relevant site. This calls into question whether E2F8 is a bona fide substrate.

Response: We agree that the R860A mutant of the c-terminal CY motif does not prevent cyclin F induced degradation of E2F8 (Fig 3A), despite its weakened interaction with cyclin F (Fig 2B). We therefore have adjusted the title of our manuscript towards E2F7. We have tested also the R408A mutant (Fig EV1F), and a R860A/R408A double mutant (data not shown), which showed weaker interaction with cyclin F, but none these CY motif mutants prevented degradation of E2F8. In contrast to E2F7, where mutation of the canonical CY motif R894A prevented cyclin F mediated degradation (Fig 3A), we were not able to identify the relevant site on E2F8 that mediates cyclin F induced degradation. It might be that the other two CY motifs at the N-terminus of E2F8 (R15A and R81A) are relevant, however mutating these sites did not weaken the interaction with cyclin F (Fig 2B). Alternatively the cyclin F mediated degradation of E2F8 might depend on additional posttranslational modification events on E2F8.

5. The ubiquitination data for E2F7 is not that convincing.

Response: We performed new ubiquitination assays for E2F7 and E2F8, and we found that E2F7 and E2F8 are strongly ubiquitinated and that cyclin F overexpression can increase the ubiquitination of E2F7/8 (Fig 3D and E). The mild increase could be related to high basal endogenous ubiquitination level for E2F7/8. We have repeated these ubiquitination experiments several times and we see consistent increase of ubiquitination of atypical E2Fs when cyclin F is overexpressed. In addition, we show that the E2F7 R894A mutant that is unable to interact with cyclin F shows reduced ubiquitination compared to the wild-type E2F7 (Fig EV2F).

6. The authors fail to fully characterize their E2F mutants, which would go a long way to making the convincing argument that these are indeed substrates. They should either show that the mutants fail to be ubiquitinated, or similarly, that they indeed have a longer half-life.

Response: We have performed new experiments to further characterize E2F7 R894A mutant. We have synchronized inducible HeLa/TO cells with hydroxyurea and then induce expression of E2F7^{WT} or E2F7^{R894A} after hydroxyurea release to compare their expression profile during cell cycle progression. Immunoblotting analysis revealed that 6 to 12 hours after addition of doxycycline the protein levels of mutant version E2F7^{R894A} were increased compared to E2F7^{WT}, whereas mRNA levels of E2F7^{R894A} were lower than E2F7^{WT} (Fig 5 And B). This finding excluded the possibility that the enhanced expression of E2F7^{R894A} was related to higher transcript levels. In addition we showed that E2F7^{R894A} mutant is less ubiquitinated than wild-type E2F7 (Fig EV2F). Then we compared the G2/M progression between the HeLa cell lines in which either E2F7^{WT} or E2F7^{R894A} were induced by doxycycline after HU release (Fig 5C). In this live cell imaging assay, over-expression of E2F7^{WT} caused a minor cell cycle delay towards mitosis (Log rank $P=0.054$), while

E2F7^{R894A} significantly reduced the number of cells to finish mitosis after 24h (Log rank $P < 0.01$). Together, these data demonstrated that expression of mutant version E2F7^{R894A} resulted in delayed G2-M progression.

7. *Why are E2F7 and E2F8 being degraded at the 12hr time point in the cell cycle experiment shown in 3D? It appears the cells are still in G2/M.*

Response: We argue that the low protein levels of E2F7/8 at 12h after release is due to an extreme low levels of RNA at this time point. We therefore performed a qPCR to test this idea, as shown in Fig EV2E. Supporting our hypothesis, RNA levels of E2F7/8 were significantly lower at 12h compared to the previous time points.

8. *The authors need to show in synchronized cells that E2F7/8 mRNA levels are not affected by Cyclin F depletion. This is a concern because 7/8 levels decrease at later time pts in the cell cycle experiment mentioned above, and also, because the mutant E2F8 still appears to be degraded.*

Response: The mRNA levels of E2F7 were not affected by cyclin F knockdown (Fig EV2E), supporting that the stabilization of E2F7 resulted from reduced proteasome degradation. E2F8 transcript levels were slightly higher at 0h and 9h and lower at 3h and 6h under cyclin F knockdown conditions compared to scr-treated cells. This finding suggests that increased transcript levels of E2F8 at 9h might contribute to the increase in protein expression of E2F8.

9. *The authors have generated a very useful allele for determining how E2F7 regulation contributes to cell cycle. The other experiments in Figures 4 and 5 are interesting, but only indicate what happens when Cyclin F is absent and all of its other (known and unknown) substrates go up. These experiments show epistasis, but don't point to a role for Cyclin F regulation of E2F7/8. Why didn't the authors express WT and non-degradable mutants and determine the effect on cell cycle.*

Response: As outlined above, we integrated wild-type and non-degradable E2F7 mutant into inducible HeLa/TO system. We found that R894A mutant of E2F7 were more stable than wild-type and it was not due to enhanced transcription (Fig 5A and B, Appendix Fig 1A shows another HeLa/TO cell clone expressing E2F7 R894A mutant). More importantly, we showed that non-degradable mutant had an enhanced effect on G2/M cell cycle progression, supporting our hypothesis that cyclin F mediated degradation of E2F7 has biological significance in the cell cycle progression.

10. *Why in the last two figures do the authors switch between using two entirely different assay systems? It makes it less convincing that the phenotypes are broadly observable.*

Response: We used two entirely different assay systems in order to be able to investigate cell cycle progression as well as DNA damage repair via live cell imaging technology. In Figure 4, we aimed to investigate the effects on cell cycle progression by employing the FUCCI system which enabled us to observe cell cycle progression in a live imaging setting. In the original Figure 5 (now as Figure 7), we aimed to investigate the dynamic of DNA damage repair and therefore RPE cells with fluorescent tagged 53BP1 was the most ideal system for this purpose.

11. It is unclear that the damage phenotypes have to do with Cyclin F regulation of 7/8 is G2. DNA damage accumulates only during DNA replication. How do the authors know that the cells are G2 arrested during the recovery?

Response: We performed a new analysis to monitor 53BP1 foci at different time point after HU release. We observed that depletion of cyclin F lead to a prolonged repair of DNA damage at 6-8h compared to scramble controls (Fig EV5C). Flow-cytometry analysis confirmed that 6-8h after HU release, cells were progressing through G2 phase of cell cycle. Moreover expression of the cyclin F non-degradable E2F7^{R894A} mutant in same time period after HU release resulted in more DNA damage (Fig EV3C), and delayed cell cycle progression compared to the cyclin F degradable wildtype version of E2F7(Fig. 5C). The findings provide further evidence that DNA damage phenotype is related to cyclin F mediated degradation of E2F7.

Minor Points

1. In figure 1, the authors show that E2F7/8 abundance is regulated by MLN4924, a small molecule that inhibits all known Cullin RING ligases (not just the SCF ligases, which are a subfamily of this much larger class). They need to clarify the language in the manuscript to reflect that the regulation of E2F7/8 by MLN4924 does not indicate SCF regulation, but instead regulation by this broader class.

Response: We have changed the manuscript accordingly.

2. Authors mention that EM11 is regulated by SCF, but only cite one of the relevant papers. Another, from the Pagano lab, should also be cited.

Response: We have add the reference in the manuscript accordingly.

3. Figure 3C should be somehow quantified since the degree of change appears minimal, especially since there is more E2F7 in the zero time pt.

Response: We have quantified the data and show here in Fig EV2C.

4. It is unclear why the authors decided to focus on Cyclin F. This should be explained.

Response: We have performed co-immunoprecipitation assay with different of F-box proteins and discovered that atypical E2Fs can bind to cyclin F. Furthermore, based on previous studies cyclin F is highly expressed during G2 when protein levels of atypical E2Fs reduce, which have been mentioned in the introduction section.

5. There is a typo in the text referring to a hydrophobic "Batch" (change to patch).

Response: This typo in the text has been corrected.

6. The authors cannot conclude from Figure 3B that the "degree" of E2F7/8 downregulation is similar to that of Cdc6. These are not quantitative assays or blots.

Response: We agree and rephrase this sentence as: "The extent of downregulation was similar to the effect of cyclin F overexpression on CDC6, a known cyclin F target".

7. In Figure 1D a loading control for another cell cycle protein should be shown, since MLN4924 is known to arrest cells in S/G2 and cause re-replication.

Response: In the new Fig 1D, we included cyclin A2 and cyclin B1 as indicators of cell cycle. While cyclin A2 slightly increase in the MLN4924 treatment, cyclin B1 was comparable between these two groups. This data suggests that cell cycle might be delayed in G2 with MLN4924

treatment, which is consistent with our observation that stabilization of atypical E2Fs occurs during G2 and contributes to a delay in cell cycle progression.

Referee #2:

Cyclin F-dependent degradation of atypical E2Fs is critical for DNA repair and G2-phase progression

The Authors propose that in addition to a previously described degradation pathway via APC/C, the atypical E2Fs (E2F7 and E2F8) are ubiquitinated by SCF-Cyclin F. This advances their turnover especially in G2 phase of the cell cycle. Deficiency in this mechanism is then linked with enhanced E2F7/8 repressor activity, which in particular has a negative impact on genome maintenance. In terms of relevance and novelty, this manuscript has merit. Firstly, Cyclin F is an essential protein with several emerging functions that remains to be fully understood. Secondly, though E2F7/8 are functionally better understood, their turnover late in G2 and early M phase was not well understood. In general, this manuscript is coherent and logical. However, considerable part of the data are not sufficiently comprehensive and they are at times also of low quality. In essence, several crucial data sets are absent in the current manuscript as outlined further below.

Major points:

1. The use of pooled siRNA needs to be validated for Cyclin F as one off-target siRNA could mediate the phenotype. Single siRNAs should be purchased and their efficacy documented. The key phenotypic analysis should be repeated with single siRNA depletion.

Response: We used individual siRNA to knockdown cyclin F and data are shown in Fig EV2B. In line with the pooled siRNA data, results from Fig EV2B demonstrated that depletion of cyclin F by single siRNA resulted in stabilization of E2F7/8 in HeLa and RPE cell lines. In addition, we performed new RNA-sequencing experiments utilizing single siRNA for cyclin F (Fig. 6).

2. The authors present data regarding the interaction between Cyclin F and E2F7/8. However, the mapping must also include Cyclin F domains to back up the ubiquitin ligase claim. Is the interaction dependent on the Cyclin domain region?

Response: To address this, we performed co-immunoprecipitations with truncated versions of cyclin F to test their interactions with EGFP-tagged E2F7/8. We found that Δ 270 truncated mutant of cyclin F, which lacks the cyclin domain, lost its binding to E2F7 while wild-type version and other truncated mutants still bond to E2F7 (Fig 2C). Cyclin F carrying a mutation in its hydrophobic patch domain (ML/AA) did not interfere with the binding to E2F7. This suggests that the interaction between cyclin F and E2F7 requires the cyclin domain but not specifically the hydrophobic patch domain of cyclin F. EGFP-E2F8 interacted with all truncated or mutants versions of cyclin F, indicating that cyclin F binds via its N-terminal domain that contains the F-box to E2F8 (Fig 2C).

3. The investigation of E2F7/8 protein levels and turnover should be carried out with significantly more scrutiny, as the presented data show a minor if any change (Figure 3). E2F8 is not a convincing target for Cyclin F. Further to Figure 3C: How can Cyclin F mediate turnover of E2F7/8 when it is absent (scr, 2h-4h-6h Chx)?

Response:

Concerning E2F8 being a target of cyclin F, we agree that the R860A mutant of the c-terminal CY motif does not prevent cyclin F induced degradation of E2F8 (Fig 3A), despite its weakened interaction with cyclin F (Fig 2B). We therefore have adjusted the title of our manuscript towards E2F7. We have tested also the R408A mutant (Fig EV1F) which showed weaker interaction with cyclin F, but none these CY motif mutants prevented degradation of E2F8. In contrast to E2F7, where mutation of the canonical CY motif R894A prevented cyclin F mediated degradation (Fig. 3A), we were not able to identify the relevant site on E2F8 that mediates Cyclin F induced degradation. It might be that the other two CY motifs at the N-terminus of E2F8 (R15A and R81A) are relevant, however mutating these sites did not weakened the interaction with cyclin F (Fig 2B). Alternatively the cyclin F mediated degradation of E2F8 might depends on additional posttranslational modification events on E2F8.

4. *The transcript data are rather weak and very biased. RNA-seq data should be included (+/-Cyclin F; +/-E2F7/8).*

Response: To determine in an unbiased manner which transcripts are regulated by atypical E2Fs in a cyclin F dependent manner, we performed RNA sequencing on nocodazole-synchronized cells treated with scr, cyclin F, *E2F7/8* or *cyclin F/E2F7/8* (triple) siRNAs. We observed a substantial overlap between genes that were downregulated by *cyclin F* siRNA compared to scr, and genes that were upregulated in *cyclin F/E2F7/8* siRNAs compared to *cyclin F* siRNAs (Fig 6A, EV4A). Gene ontology analysis showed that these genes, which were downregulated after cyclin F knockdown, and rescued by additional E2F7/8 knockdown, were strongly enriched for DNA repair and replication pathways (Fig 6B, Fig EV4B, also shown in Table 2. DEgenes_RNaseq_CCNF). Among these DNA repair genes, we observed many known E2F7/8 target genes, such as *RAD51*, *MSH2/6*, *EXO1*, and *CHEK1*. Quantitative PCR on a subset of these DNA repair genes confirmed that they were indeed downregulated by cyclin F depletion in an E2F7/8-dependent manner (Fig 6C).

5. *Evidence for changes in protein levels is crucial for key factors (RAD51, and others identified in point 4).*

Response: We measured the protein levels of Rad51 and Chk1, two crucial factors for DNA damage repair. As shown in the Fig 6D, *sicyclin F* decreased the protein levels of Rad51 and Chk1, and their expressions were rescued by additional knockdown of E2F7/8. This data demonstrated that cyclin F regulation of DNA repair factors had effect on their transcription and protein level.

6. *Repair deficiency must be validated, are cell systems (those mentioned in above point 4.) HRR-deficient or are other repair pathways involved?*

Response: It is intriguing that multiple DNA damage repair pathways, including mismatch repair (MMR), nucleotide excision repair (NER), base excision repair (BER) and homologous recombination (HR) are regulated by atypical E2Fs in a cyclin F dependent manner (Fig 6B). To validate the repair deficiency of homologous recombination, we used a HeLa cell lines which stably transformed with pDR-GFP construct, a system enabled us to monitor the homologous recombination efficiency by measuring the GFP abundance. We then transfected them with siRNA against cyclin F, and we found that depletion of cyclin F significantly reduced HR repair efficiency compared to scramble control. This repair deficiency was fully recovered by additionally depletion of E2F7/8 (Fig 7A and Fig EV 5A), suggesting that this HR deficiency induced by cyclin F knockdown is dependent on E2F7/8.

Additional points:

1. *Abstract claim in last sentence "...essential for DNA repair and cell cycle progression" is an overstatement.*

Response: We have rephrased this sentence in the abstract, now stating: "plays an important role in DNA repair and cell cycle progression"

2. *Inputs vary (Fig. 2C, 3A), there is a need to repeat and replace. Many of the immunoblot panels appear to be substantially image processed, this greatly impacts figure quality and must be addressed (Fig. 1A, 1C, 1D, 2A, 3B, 3D, 3E EV1B, EV2C). The authors also should take care to present data without nocodazole (could be supplemental).*

Response: We agree that the inputs varied among the samples in original Fig 2C and 3A, but these inputs (EGFP in Fig2C, and Flag in Fig 3A) did not have an impact on the interpretation of the results. We did not process the images and the reason for relatively low figure quality is related to the low quality of the antibodies against endogenous cyclin F, E2F7 and E2F8 . However these antibodies are very specific as demonstrated by our siRNA and CRISPR/Cas9 experiments(Fig. EV2B, EV3A).

3. *Figure 4: using 16h HU for synchronizing might not be the method of choice for CKNF-KD. Aphidicolin is a better choice.*

Response: We agree to this point that there are multiple methods for cell cycle synchronization, but the main reason of using HU was not only to synchronize cell cycle but also to induce DNA damage. In the original Fig 5C and D (now as Fig 7), we wanted to study also the DNA damage repair recovery.

4. *Figure 4B+D: Could rather be presented in %. (same scale, but easier to grasp)*

Response: We have changed the figures accordingly.

5. *Foci number cannot be presented with two decimals.*

Response: We removed decimals behind the comma in Fig EV5B.

6. *Text links to EV2E and EV2F needs correction.*

Response: We corrected the text links to EV2E and EV2F.

7. *Replication fork restart processes may contribute to phenotypes, this should be mentioned.*

Response: We are happy to make textual adjustments on the request of the reviewer, but we are not exactly sure what the reviewer is referring to. Is it related to the use of hydroxyurea?

Referee #3:

In this manuscript the authors have investigated the regulation of the atypical E2Fs 7 and 8 during G2 phase. They find that E2F7 and 8 are targets of Cyclin F-mediated degradation and identify the Cyclin F binding domains. Further, they show that loss of E2F7/8 rescues cell cycle defects by

Cyclin F degradation. They also find that multiple known Cyclin F targets, involved in DNA replication and repair, are affected by E2F7/8 loss and propose that E2F7/8 may function as an inherent compensation mechanism in case of reduced Cyclin F function.

In general, the manuscript is well written and the study is well performed. The identification of E2F7/8 as Cyclin F targets has clear implications for genome stability and cell cycle control. However, I have three main objections that I think should be addressed.

First, quite some of the measurements are performed in Nocodazole arrested cells, with the reasoning that it is important to compare cells in similar cell cycle phases. While I completely agree to this reasoning, depending on cell type and concentration, Nocodazole typically blocks most cells in prometaphase (not prophase), but can also as a stress-response block cells in G2. As the various treatments (in Fig 5 and others) impact the G2/M transition, it is important to show the distribution of G2 vs M in Nocodazole treated samples after E2F7/8 and Cyclin F RNAi.

Response: To show the distribution of G2 vs M in nocodazole treatment, we performed MPM staining to determine the percentage of mitotic cells by flow cytometry analysis (in the new Fig EV4E). We found that in all siRNA conditions, more than 40% of the cells were arrested in M phase. This data demonstrate that cell cycle phases were comparable among all the groups targeting cyclin F/E2F7/8 under nocodazole treatment.

Second, large parts of the reasoning focus on that Cyclin F-E2F7/8 impact on G2 progression through regulation of DNA repair. Whereas I agree that the data supports this as a contributing factor, the relative importance versus repression of cell cycle promoting factors as Cdk1, Cyclin B or Plk1 is not clear. This should be discussed.

Response: Interestingly, we found that genes, known to control mitotic activity, such as *PLK1* (polo-like kinase 1) and *CCNB1* (cyclin B1) were upregulated in response to *cyclin F* knockdown (Fig EV4D), but are not affected by the knockdown of atypical E2Fs. This is in line with a previous study where it has been shown that cyclin F suppresses a B-Myb driven transcriptional program regulating mitotic gene expression (Klein DK et al. *Cyclin F suppresses B-Myb activity to promote cell cycle checkpoint control. Nat Commun.* 2015). Phosphorylated MPM2 as an indicator of M phase was slightly increased in *sicyclin F* and siTriple conditions compared to scr and siE2F7+8 conditions, suggesting that depletion of cyclin F also affects mitotic entry through regulation of B-Myb target gene expression (Fig EV4E). These results and interpretation are included in the revised version of our manuscript.

Third, 53BP1 binding to damaged DNA is counteracted by mitotic kinases - which are gradually activated through G2 phase - and the absence of 53BP1 foci in cells close to mitosis cannot be taken for a guarantee that damaged DNA is repaired. Line 303-305, it should be specified that these are cells containing DNA damage due to HU treatment. Line 281-282, to make this point a mitotic kinase-independent measurement of DNA damage is required.

Response: In addition to the 53BP1 assay, we include an new analysis to measure DNA damage repair independent of mitotic kinase, which is the Homologous Recombination (HR) assay. As shown in the Fig 7A and Fig EV5A, *sicyclin F* induced an HR repair deficiency and this phenotype was rescued by additional knockdown of E2F7/8. Taken together with the data from Fig 7B, which showed that accumulation of γ -H2AX was mediated by cyclin F/E2F7/8, we demonstrate that degradation of atypical E2Fs by cyclin F plays an important role in DNA repair.

Minor points:

Lane 319, I don't find this figure.

Response: The original sentence here was to discuss whether cyclin F was down-regulated under DNA damage response in an ATR-dependent manner. To this end, we performed a new experiment in which we synchronized HeLa cells with either Nocodazole or HU (Appendix Fig 1B). We did observe that cyclin F was down-regulated with the treatment of Nocodazole and HU. However, we cannot exclude that this decrease of cyclin F was due to cell cycle synchronization. We realized that this discussion could be distracting to the reader so we decided to remove this sentence from the discussion part.

Figure 4F - it's unclear to me whether this looks at cells that did not complete mitosis (lane 228) or the whole population (lane 651). I think that the addition of a (EV) figure similar to 5C showing the durations of all cells in fig 4C-F would help the interpretation.

Response: In this assay, we quantify only the cells that did not complete mitosis. We have changed the text in the figure legend accordingly to avoid misunderstanding.

I'd suggest a visual schematic figure for the proposed role of Cyclin F and E2F7/8 on target gene expression.

Response: We have included a schematic figure as the new Fig 8.

It would be good if asterisks on blots were used consistently - for example in fig1 it indicates specific band, in Ev1 unspecific band, and in some E2F7 stainings with multiple bands no asterisk is included.

Response: We have corrected this issue. Now all the asterisks consistently indicates specific bands in all blots.

The reference list contains many doublets.

Response: We have changed the manuscript accordingly.

Appendix Fig 1 for Referees.

Figure Legend

- A. Disruption of cyclin F binding site increased stability of E2F7. E2F7^{WT} and E2F7^{R894A} constructs were integrated in to HeLa/TO system (second cell line clone to the experiment Fig 5A). Cells were arrested with HU for 16 hours before releasing into doxycycline containing medium, and cells were collected every 3 hours for immunoblotting.
- B. Nododazole and Hydroxyurea treatment down-regulated expression of cyclin F. HeLa cells were treated with 16 hours of Nododazole or Hydroxyurea before harvesting for immunoblotting.

2nd Editorial Decision

31st July 2019

Thank you for submitting your revised manuscript for our consideration. It has now been assessed again by referee 2, who appears generally satisfied with the revision and only retains a few minor comments that I would ask you to incorporate into the text of a final, re-revised version. After this, we shall be happy to accept the study for publication in The EMBO Journal.

In addition, there are several editorial issues to address during the final minor revision.

 REFEREE REPORTS

Referee #2:

The authors have substantially improved the manuscript with additional molecular mechanistic (mainly the interaction with Cyclin F) and RNAseq data.

It is very intriguing that Cyclin F suppresses E2F7(8), yet, Cyclin F also limits cell cycle driving B-Myb and E2F1-3 activities (papers by Klein et al and Clijsters et al). The authors should further elaborate (in the discussion) on the cell cycle impact of their findings in relation to these publications.

2nd Revision - authors' response

10th August 2019

Referee #2:

The authors have substantially improved the manuscript with additional molecular mechanistic (mainly the interaction with Cyclin F) and RNAseq data.

It is very intriguing that Cyclin F suppresses E2F7(8), yet, Cyclin F also limits cell cycle driving B-Myb and E2F1-3 activities (papers by Klein et al and Clijsters et al). The authors should further elaborate (in the discussion) on the cell cycle impact of their findings in relation to these publications.

Response: We have further elaborated these findings in the discussion part of our revised manuscript.

Accepted

13th August 2019

Thank you for submitting your final revised manuscript for our consideration. I am pleased to inform you that we have now accepted it for publication in The EMBO Journal.

Corresponding Author Name: Alain de Bruin

Journal Submitted to: EMBO JOURNAL

Manuscript Number: EMBOJ-2018-101430